# Stapling strategy for slowing helicity interconversion of α-helical peptides and isolating chiral auxiliary-free one-handed forms

Naoki Ousaka [1] ✉, Mark J. MacLachlan [1,2,3] ✉ & Shigehisa Akine [1,4] ✉

In nature, α-helical peptides adopt right-handed conformations that are dictated by L-amino acids. Isolating one-handed α-helical peptides composed of only achiral components remains a significant challenge. Here, this goal is achieved by optical resolution of the corresponding racemic (quasi-)static α-helical peptide with double stapling, which effectively freezes the interconversion between the right-handed (*P*)- and left-handed (*M*)-α-helices. An as-obtained doubly stapled analogue having an unprotected L-valine residue at the C-terminus transforms from a kinetically trapped (*M*)-α-helix to a thermodynamically stable (*P*)-α-helix upon heating. In contrast, the corresponding singly stapled α-helical peptide undergoes an acid/base-triggered and solvent-induced reversible inversion of its preferred helicity within minutes. The interconversion rates of the singly and doubly stapled α-helical peptide foldamers are approximately $10^6$ and $10^{12}$ times slower, respectively, than that of a non-stapled dynamic helical peptide. Therefore, the enantiopure doubly-stapled (quasi-)static α-helical peptide would retain its optical activity for several years at 25 °C.

Since the discovery of the α-helix in 1951[1], a wide variety of synthetic helical oligomers (foldamers)[2–5], polymers[6–9], and supramolecules[9–11] composed of abiotic backbones have been synthesized to mimic biological helical structures and to gain new functions that cannot be achieved using biopolymers[9]. Most of the helices, including the biological polymers, exclusively adopt either a right-handed (*P*) or left-handed (*M*) helical conformation that is dictated by their chiral components, such as L-amino acids for the (*P*)-α-helix and triple-stranded (*M*)-helical collagen, and D-sugars for (*P*)-double helical DNA. On the other hand, artificial helical polymers and oligomers consisting only of achiral components form racemic mixtures of both (*P*)- and (*M*)-helices in a 1:1 ratio. If the helices are sufficiently dynamic to allow *P/M*

interconversion[7,8], a preferred helical sense can be easily induced under thermodynamic control through covalent or noncovalent interactions of chiral auxiliaries at the chain end[12–17] or side chains[6–9]. In this case, however, removal of the chiral auxiliaries leads them to racemize back to the original optically inactive helices. In contrast, if there is a sufficiently high energy barrier for the *P/M* interconversion, the one-handed helices are static and no racemization takes place. Indeed, optically active helices have been obtained through optical resolution of racemic static helices[18,19] including spontaneous resolution during crystallization[20] or helix sense-selective synthesis[7,21], even if they contain no chiral auxiliaries. Such an excess one-handed helix can also be prepared by the chiral memory strategy, e.g., induction of a

[1]Nano Life Science Institute (WPI-NanoLSI), Kanazawa University, Kakuma-machi, Kanazawa 920-1192, Japan. [2]Department of Chemistry, University of British Columbia, 2036 Main Mall, Vancouver, BC V6T 1Z1, Canada. [3]Quantum Matter Institute, University of British Columbia, 2355 East Mall, Vancouver, BC V6T 1Z4, Canada. [4]Graduate School of Natural Science and Technology, Kanazawa University, Kanazawa 920-1192, Japan. ✉e-mail: ousaka@staff.kanazawa-u.ac.jp; mmaclach@chem.ubc.ca; akine@se.kanazawa-u.ac.jp

preferred helix sense in dynamically racemic helices through non-covalent interactions with optically active additives followed by their transformation into optically active static helices via removal of the chiral additives[22–25].

Similar to the dynamic abiotic helices, achiral peptide chains composed mainly of strongly helicogenic $C^\alpha$-tetrasubstituted achiral amino acids[26,27] are known to fold into a dynamic helical structure. Detailed conformational analysis has shown that they adopt a $3_{10}$-helical conformation[28] characterized by 3 amino acids per turn and intramolecular hydrogen bonds between residues at the $i$ (C = O) and $i + 3$ (NH) positions to form 10-membered hydrogen-bonded rings rather than the well-known $\alpha$-helical one ($3.6_{13}$-helix) with hydrogen bonds between residues at the $i$ (C = O) and $i + 4$ (NH) positions. These $3_{10}$-helices are sufficiently dynamic to rapidly interconvert between the ($P$) and ($M$) forms on the millisecond time scale[29,30]. However, it is noteworthy that single stapling of such a dynamically racemic $3_{10}$-helical peptide effectively decelerates the $P/M$ interconversion[31]. The stapling strategy has also been used not only to stabilize the $\alpha$-helical structures along with improvement of proteolytic stability and cell permeability[32,33] but also to suppress the $P/M$ interconversion of helical foldamers[34,35]. However, even though the $\alpha$-helix is a ubiquitous and indispensable structural motif in living systems, one-handed static or well-defined dynamic $\alpha$-helical peptides composed only of achiral building blocks have never been constructed.

Here, we report one-handed static $\alpha$-helical oligopeptides consisting only of achiral components, whose racemization is effectively prevented by double stapling of the peptides. The singly stapled dynamic $\alpha$-helical peptides exhibit slow $P/M$ interconversion on a time scale of minutes at room temperature. These kinetics were estimated by using the simple acid/base-triggered reversible helicity inversion system, because the excess one-handed forms without any chiral auxiliary would undergo racemization on the chromatographic time scale. A key to this system is the introduction of an unprotected L-Val-OH residue as a switchable helicity controller at the C-terminus of the singly stapled achiral peptide chain, whose preferred helix sense is shown to be reversibly switched under thermodynamic control by the addition of a suitable acid or base or by changing solvents. In contrast, a kinetically trapped helix sense was found for an as-obtained form of the corresponding doubly stapled peptide, which can be transformed into a thermodynamically stable form with opposite helix sense by thermal annealing at high temperature. Therefore, the second stapling of the dynamic $\alpha$-helical peptides freezes the $P/M$ interconversion at room temperature, which then takes place on a time scale of years, thus producing the enantiopure doubly stapled static $\alpha$-helical peptides following optical resolution using chiral chromatography (Fig. 1a).

## Results

### Molecular design and synthesis of stapled peptides

The preferential formation of the $3_{10}$- or $\alpha$-helix is highly dependent on the sequences of helicogenic achiral amino acid residues[36], such as $\alpha$-aminoisobutyric acid (Aib)[37], 1-aminocyclohexane-1-carboxylic acid (Ac$_6$c)[38] and 4-aminopiperidine-4-carboxylic acid (Api)[39], external stimuli (temperature[40] and solvent[39,41]) and aggregation[42], although most of the dynamic helical peptides have been found to adopt the $3_{10}$-helical conformation rather than an $\alpha$-helix. For instance, homo-oligopeptides of the Aib residue, -(Aib)$_n$- ($n = 3$–10), have been found to adopt only the $3_{10}$-helical conformation[26], whereas oligopeptides containing a more bulky -(Ac$_6$c)$_n$- segment ($n = 4$ or 6) adopt the $3_{10}$- and/or $\alpha$-helix, depending on solvent polarity[43,44]. This increased $\alpha$-helix propensity is likely because the bulky cyclohexyl side chains in the oligomeric Ac$_6$c segments prefer to adopt a staggered arrangement along the helical axis to avoid steric hindrance between the side chains, thereby inducing the $\alpha$-helical conformation in the backbone. Therefore, we chose the Ac$_6$c residue and its piperidine analog, Api

residue, in order to build achiral peptide chains with the potential to adopt the $\alpha$-helical conformation; the Api residue has a piperidine side-chain available for an intramolecular cross-linking reaction, i.e., stapling.

For single stapling of $\alpha$-helical peptides, intramolecular cross-linking is usually performed between two side chains of amino acid residues at $i$ and $i + 4$, $i + 7$, or $i + 11$ positions[32,33,45,46]. These positions are located on the same face of the $\alpha$-helix, whereas in the $3_{10}$-helix they are much farther apart. Therefore, if a cross-linker of appropriate length is selected, single stapling may exclusively induce the $\alpha$-helical conformation in the peptide backbones (Supplementary Fig. 5a). Our aim in this study is to construct stapled $\alpha$-helical peptides with almost no interconversion between the ($P$) and ($M$) forms. Stapling with a rigid cross-linker may increase the energy barrier for the $P/M$ interconversion of the dynamic $\alpha$-helical peptides by destabilizing non-$\alpha$-helical conformations including intermediates of the helicity inversion process. Our modeling study suggested that a rigid cross-linker based on biphenyl-4,4'-diacetic acid would be suitable for the cross-linking between the side chains of the Api residues incorporated at $i$ and $i + 7$ positions of Ac$_6$c-based achiral peptide chains. We also chose $i + 2$ and $i + 9$ positions of the achiral peptides for the additional stapling, in order to avoid undesired stapling, e.g., the cross-linking between the $i + 1$ and $i + 7$ positions may occur due to the $3_{10}$-helix formation when the Api residues for the second stapling are introduced at $i + 1$ and $i + 8$ positions (Supplementary Fig. 5b).

We anticipated that the C-terminal unprotected L-Val-OH residue covalently introduced at the C-terminus of the stapled achiral peptides would act as a switchable helicity controller, in which a preferred helix sense of the stapled dynamic $\alpha$-helical peptides could be switched by acid/base-triggered reversible protonation/deprotonation of the C-terminal carboxy group (Fig. 1b)[17,47]. This simple switching system may facilitate monitoring of the kinetics of the $P/M$ interconversion of the stapled dynamic $\alpha$-helical peptides by circular dichroism (CD) measurements.

Based on the molecular design strategies discussed above, we synthesized the singly stapled decapeptide **c1-Val-OH** and the doubly stapled dodecapeptide **dc2-Val-OH**, both of which have unprotected L-Val-OH residues at the C-terminus (Fig. 1c). The doubly stapled dodecapeptide **dc2-Aib-OMe** consisting only of achiral components was also synthesized to obtain the one-handed static $\alpha$-helical peptides via optical resolution. In addition, **c1-Val-O$^t$Bu** with the C-terminal-protected L-Val-O$^t$Bu ($^t$Bu = *tert*-butyl) residue was prepared for comparison with **c1-Val-OH**. The cross-linking reactions between the free piperidine side chains and an activated diester cross-linking reagent were carried out under dilute conditions after the stepwise synthesis of the peptide backbones. All the peptides were characterized by nuclear magnetic resonance (NMR) spectroscopy and electrospray ionization time-of-flight (ESI-TOF) mass spectrometry, and their purities were confirmed by high-performance liquid chromatography (HPLC) analyses (for synthesis and characterization, see the Supplementary Information).

### Conformational analyses of singly and doubly stapled peptides

The CD spectrum of the C-terminal-protected **c1-Val-O$^t$Bu** in CH$_2$Cl$_2$ displayed an intense positive Cotton effect at 224 nm reflecting an ($M$)-helical conformation[48], suggesting that **c1-Val-O$^t$Bu** adopted a $3_{10}$- or $\alpha$-helical conformation with an excess of the ($M$)-handedness (Fig. 2a (iv) and Supplementary Fig. 21a). The negative CD signal around 260 nm is due to the biphenyl chromophore whose axially twisted conformation with an excess of a one-handed twist sense is correlated with the preferred helix sense of the peptide main chain. This assignment of the ($M$)-helix preference was also supported by the positive CD signals at 208 and 224 nm in protic solvents (methanol (MeOH) and 2,2,2-trifluoroethanol (TFE)) and at 224 nm in CH$_2$Cl$_2$ (Supplementary Fig. 25a(i), (ii)). It is well-known that typical ($M$)-$\alpha$-helical peptides

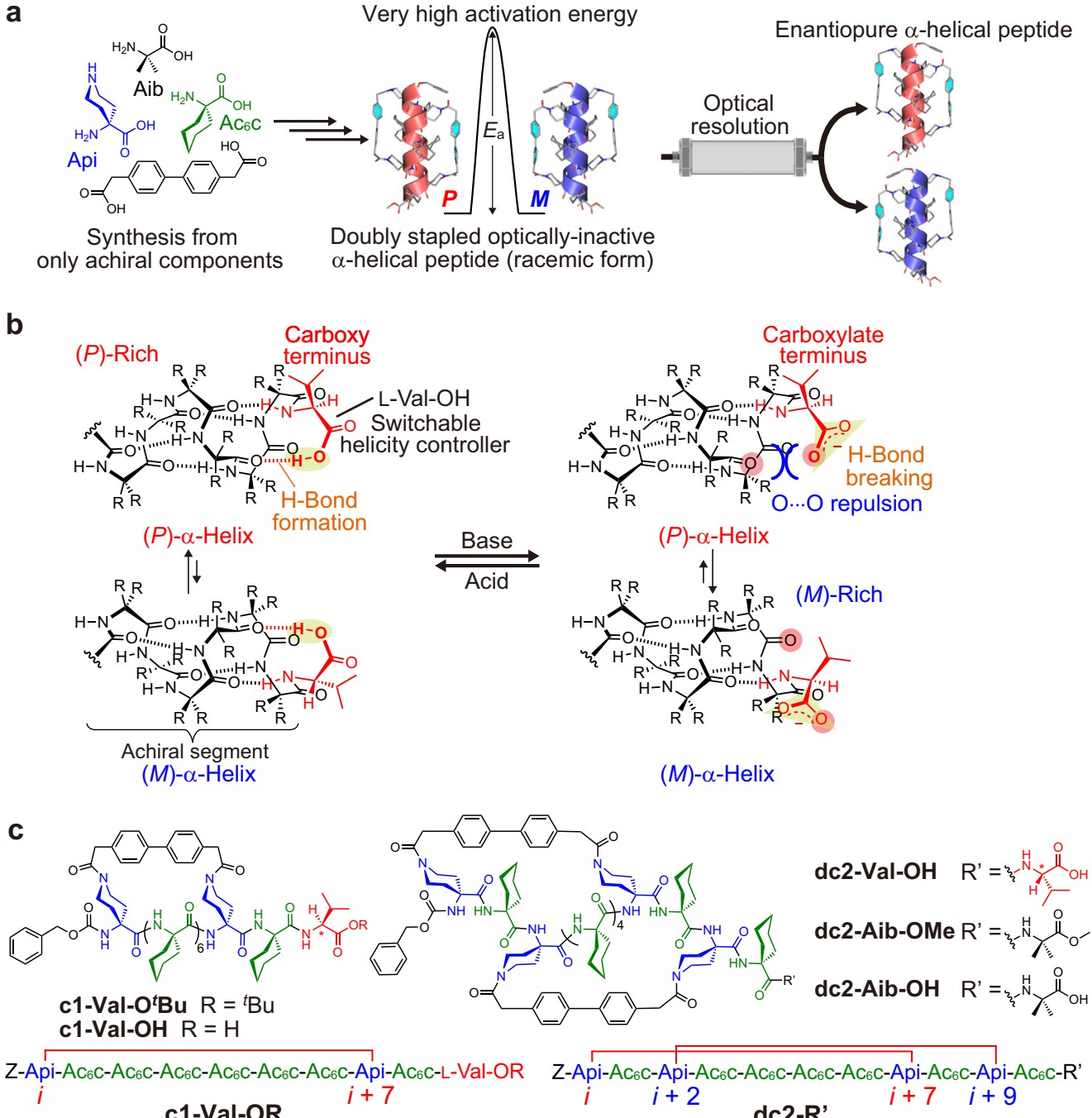

**Fig. 1 | Singly and doubly stapled dynamic or (quasi-)static α-helical peptides.**
**a** Schematic representation of the strategy to construct a one-handed static α-helical peptide from only achiral components. The doubly stapled peptide has a very high activation energy ($E_a$) for interconversion between (*P*)- and (*M*)-α-helices (racemization), enabling its optical resolution by chiral chromatography.
**b** Schematic representation of the acid/base-triggered reversible helicity inversion of a dynamic α-helical peptide possessing the unprotected L-Val-OH residue at the C-terminus by reversible formation/breaking of an intramolecular hydrogen bond at the C-terminal region as a result of protonation/deprotonation of the C-terminal carboxy group. The L-Val-OH residue introduced at the C-terminus of the achiral peptides acts as a switchable helicity controller. **c** Chemical structures of the singly stapled helical peptides, **c1-Val-O^tBu** and **c1-Val-OH**, and the doubly stapled helical peptides, **dc2-Val-OH**, **dc2-Aib-OMe** and **dc2-Aib-OH**. The amino acid sequences of these stapled peptides are also shown.

---

exhibit a CD spectral pattern with two positive maxima at 208 and 222 nm of similar intensity. A similar CD spectral pattern was observed for **c1-Val-O^tBu** in MeOH and TFE, although its CD intensities at 222 nm were more intense than those at 208 nm. This intense signal is most likely due to the contribution of the biphenyl chromophore. In the ¹H NMR spectrum of **c1-Val-O^tBu** in CD₂Cl₂ at 298 K, two sets of signals were observed due to slow interconversion between the diastereomeric (*P*)- and (*M*)-helices on the NMR time scale, and the *P/M* molar

ratio was 38/62 (Supplementary Fig. 21c). The C-terminal L-amino acid ester is known to induce an (*M*)-helix preference in dynamic 3₁₀-helical peptides because of O···O repulsion between an oxygen atom of the ester functional group and the carbonyl oxygen atom of the third amino acid residue from the C-terminus[43,49]. In contrast to this (*M*)-helix preference, interestingly, the C-terminal unprotected **c1-Val-OH** adopted an excess of the (*P*)-helical conformation with the *P/M* molar ratio of 78/22 in CD₂Cl₂ at 298 K, as revealed by the CD and ¹H NMR

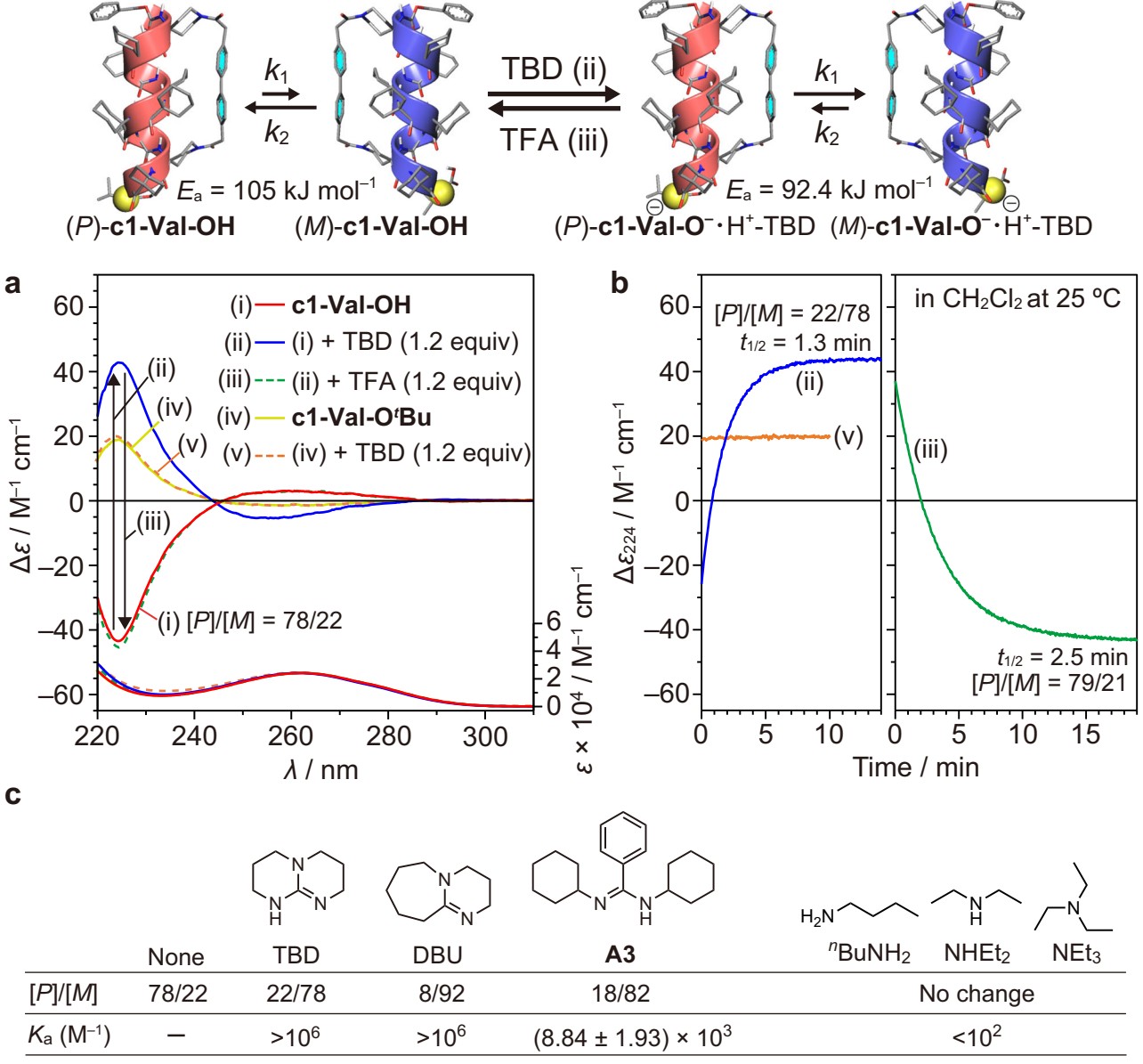

**Fig. 2 | Acid/base-triggered reversible helicity inversion of c1-Val-OH monitored by circular dichroism (CD) spectroscopy. a** CD (top) and absorption (bottom) spectra (CH$_2$Cl$_2$, 0.50–0.45 mM, 25 °C) of **c1-Val-OH** (i), (i) + TBD (1.2 equiv) (ii), (ii) + TFA (1.2 equiv) (iii), **c1-Val-O$^t$Bu** (iv) and (iv) + TBD (1.2 equiv) (v) at the thermodynamic equilibrium state. **b** Time-dependent CD intensity changes at 224 nm ($\Delta\varepsilon_{224}$) for **c1-Val-OH** + TBD (1.2 equiv) (ii) and (ii) + TFA (1.2 equiv) (iii) and for **c1-Val-O$^t$Bu** + TBD (1.2 equiv) (v) in CH$_2$Cl$_2$ at 25 °C. The CD intensity changes

were monitored starting immediately after the addition of TBD or TFA. For detailed experimental procedures, see Methods. **c** Summary of the molar ratios of ($P$) and ($M$)-**c1-Val-OH** ([$P$]/[$M$]) in the presence of different organic bases in CH$_2$Cl$_2$ at the thermodynamic equilibrium state at 25 °C and the association constants ($K_a$, M$^{-1}$) of **c1-Val-OH** with various bases. For the determination of the [$P$]/[$M$] molar ratios and $K_a$ values, see Methods and the Supplementary Information.

measurements (Fig. 2a(i) and Supplementary Fig. 21a, b). This ($P$)-helix preference is most likely due to the intramolecular hydrogen bond formation between the C-terminal carboxy proton of the L-Val-OH residue and a carbonyl oxygen atom of the third or fourth amino acid residue from the C-terminus (Fig. 1b). Such a hydrogen-bonding pattern has been observed for the reported C-terminal unprotected 3$_{10}$-helical Z-(Aib)$_{10}$-OH (Z = benzyloxycarbonyl) in the solid state[50]. In addition, a similar ($P$)-helix preference has also been found in the reported dynamic 3$_{10}$-helical peptides having a C-terminal L-amino acid residue capped with a primary amide group, in which the C-terminal amide NH proton forms an intramolecular hydrogen bond in a manner similar to the carboxy proton of the C-terminal unprotected peptides[43,44,51]. However, the preferred helix sense of **c1-Val-OH** in MeOH was found to be opposite to that in CH$_2$Cl$_2$ and TFE (Fig. 2a(i)

and Supplementary Fig. 25a (iii), (iv)), although the CD spectral patterns of **c1-Val-OH** in these protic solvents were similar to those of **c1-Val-O$^t$Bu**, except for the CD sign. This methanol-induced helicity inversion is mainly due to disruption of the intramolecular hydrogen-bonding of the terminal carboxy proton.

A density functional theory (DFT) study revealed the α-helix formation of **c1-Val-OH**; the average dihedral angles (|$\phi$|/|$\psi$| = 55°/44°) in the achiral peptide segment were very close to those of the typical α-helix (57°/47°)[52], although the N-terminal Api(1) residue (the number in parentheses represents the residue number from the N-terminus) seemed to adopt a 3$_{10}$-helix-like conformation to maximize the number of the intramolecular hydrogen bonds at the N-terminal region (Fig. 3a and Supplementary Tables 1 and 2). Moreover, this stapled α-helical structure is 50.4 kJ mol$^{-1}$ more stable

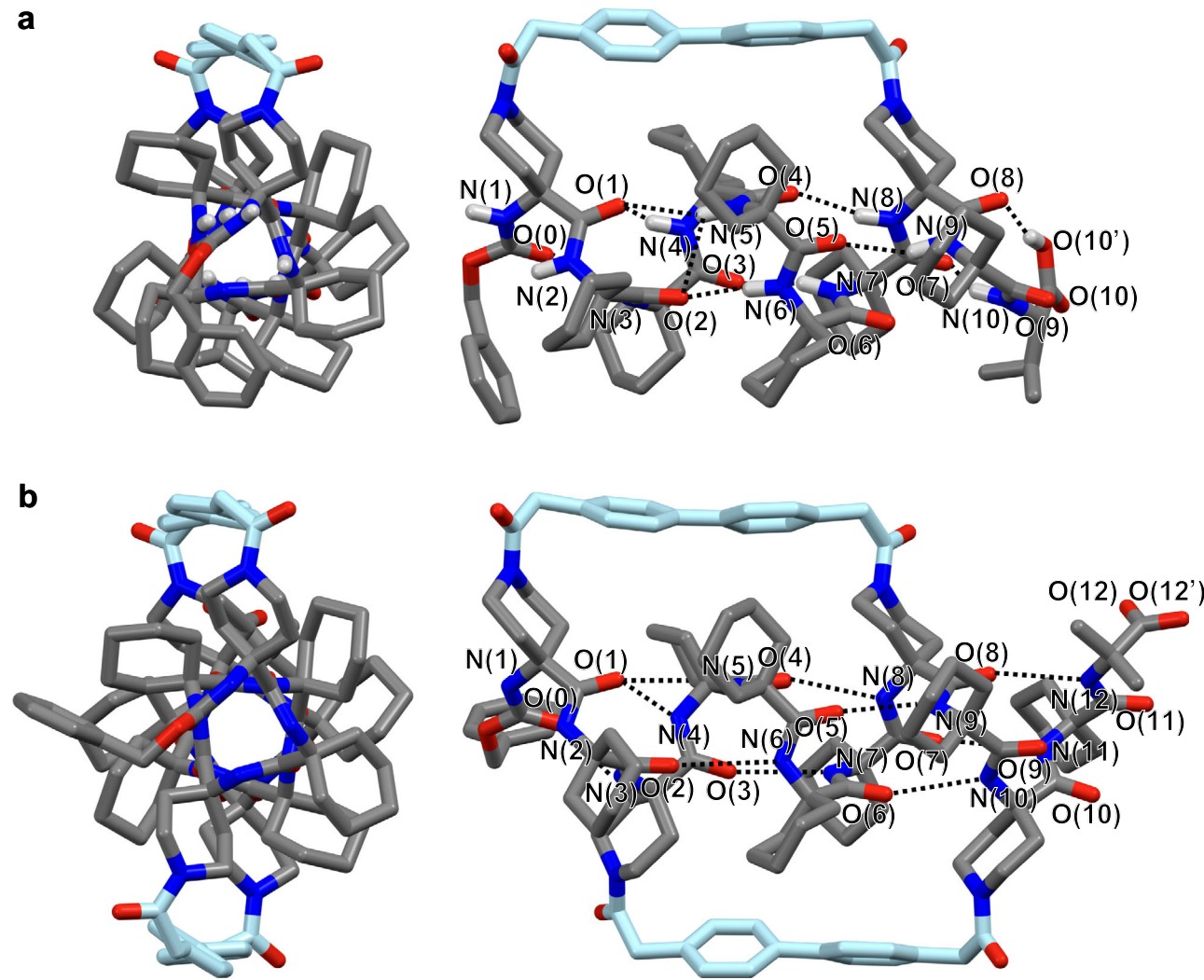

**Fig. 3 | Stapled α-helical structures of c1-Val-OH and dc2-Aib-OH. a** The energy-minimized structure of (*P*)-α-helical **c1-Val-OH** obtained by DFT calculation (top (left) and side (right) views). All of the hydrogen atoms except for the amide and C-terminal carboxy protons are omitted for clarity. **b** The molecular structure of (*P*)-α-helical **dc2-Aib-OH** as determined by single-crystal X-ray diffraction (top (left) and side (right) views). Only the (*P*)-helix is shown. All of the hydrogen atoms, minor disordered parts, and solvent molecules are omitted for clarity. The dihedral angles (*ϕ*, *ψ*, and *ω*) and hydrogen-bonding parameters of **c1-Val-OH** and **dc2-Aib-OH** are summarized in Supplementary Tables 1, 2, 10, and 11, respectively. The number in parentheses represents the residue number from the N-terminus.

than the $3_{10}$-helical structure with the average dihedral angles ($|\phi|/|\psi| = 55°/26°$) due to the unfavorable structural constraint imposed by the stapling (Supplementary Fig. 8 and Supplementary Tables 3 and 4). The structural features seen in the DFT-calculated α-helical structure were consistent with the following NMR measurement results in solution: (i) the two-dimensional (2D) nuclear Overhauser effect spectroscopy (NOESY) of **c1-Val-OH** showed strong NOEs between the adjacent N(*i*)H–N(*i* + 1)H signals, except for the overlapping signals; (ii) one of the side-chain cyclohexyl protons at each of the $\text{Ac}_6\text{c}(4)$ and $\text{Ac}_6\text{c}(5)$ residues showed upfield shifts compared to the others, due to the ring current effect of the biphenyl moiety (Supplementary Figs. 12–15); and (iii) the amide N(1)H and N(2)H protons did not form the intramolecular hydrogen bond and were therefore exposed to the solvent, as indicated by their high temperature coefficients obtained from variable-temperature (VT) $^1$H NMR measurements (Supplementary Fig. 16).

Similar to the singly stapled α-helical **c1-Val-OH**, the 2D-NOESY and VT $^1$H NMR measurement results of the doubly stapled **dc2-Aib-OMe** were also in good agreement with the DFT-calculated α-helical structure with the average $|\phi|/|\psi|$ dihedral angles of 55°/45° (Supplementary Figs. 9a and 17–20 and Supplementary Tables 5

and 6). Based on the DFT calculations, the α-helical structure is 80.5 kJ mol$^{-1}$ more stable than the $3_{10}$-helix, clearly indicating that the second stapling further destabilized the $3_{10}$-helix (Supplementary Fig. 9). Fortunately, single crystals of the doubly stapled, C-terminal unprotected **dc2-Aib-OH** (Fig. 1c) suitable for an X-ray diffraction study were obtained by slow evaporation of a solution of **dc2-Aib-OMe** in a CDCl$_3$/(CD$_3$)$_2$SO mixture (~20/1, v/v) via unexpected hydrolysis of the C-terminal methyl ester group. The structure of **dc2-Aib-OH** determined by single-crystal X-ray diffraction revealed that it adopts a typical α-helical conformation (average $|\phi|/|\psi| = 56°/48°$), with two stapling biphenyl linkages as initially designed. In contrast, the C-terminal Aib-OH residue was found to adopt three different conformations (components A, B and C) due to disorder (Fig. 3b and Supplementary Fig. 11 and Supplementary Tables 10 and 11); the terminal carboxy group of components A and B formed an intermolecular hydrogen bond with a co-crystallized dimethyl sulfoxide molecule, whereas that of disordered component C formed an intramolecular hydrogen bond with the carbonyl oxygen atom of the $\text{Ac}_6\text{c}(9)$ residue (Supplementary Fig. 11). The α-helical structure observed in the solid state is consistent with the DFT-calculated structure and the NMR results measured in solution.

**Table 1 | Rate constants ($k_1$ and $k_2$, sec$^{-1}$) and half-life time ($t_{1/2}$, min) for the interconversion between ($P$)- and ($M$)-c1-Val-O$^-$·H$^+$-TBD and between ($P$)- and ($M$)-c1-Val-OH[a]**

| Additive | Temp (°C) | [$P$]/[$M$] | $k_1 + k_2$ (sec$^{-1}$)[d] | $k_1$ (sec$^{-1}$)[e] | $k_2$ (sec$^{-1}$)[e] | $t_{1/2}$ (min)[f] |
|---|---|---|---|---|---|---|
| TBD | 35 | 22/78[b] | $2.91 \times 10^{-2}$ | $2.27 \times 10^{-2}$ | $6.41 \times 10^{-3}$ | 0.4 |
| TBD | 25 | 22/78[c] | $8.79 \times 10^{-3}$ | $6.86 \times 10^{-3}$ | $1.93 \times 10^{-3}$ | 1.3 |
| TBD | 15 | 22/78[b] | $2.45 \times 10^{-3}$ | $1.91 \times 10^{-3}$ | $5.39 \times 10^{-4}$ | 4.7 |
| TBD | 5 | 22/78[b] | $5.91 \times 10^{-4}$ | $4.61 \times 10^{-4}$ | $1.30 \times 10^{-4}$ | 19.6 |
| TFA + TBD | 35 | 79/21[b] | $1.86 \times 10^{-2}$ | $3.91 \times 10^{-3}$ | $1.47 \times 10^{-2}$ | 0.6 |
| TFA + TBD | 25 | 79/21[c] | $4.58 \times 10^{-3}$ | $9.61 \times 10^{-4}$ | $3.61 \times 10^{-3}$ | 2.5 |
| TFA + TBD | 15 | 79/21[b] | $8.40 \times 10^{-4}$ | $1.76 \times 10^{-4}$ | $6.64 \times 10^{-4}$ | 13.8 |
| TFA + TBD | 5 | 79/21[b] | $2.37 \times 10^{-4}$ | $4.97 \times 10^{-5}$ | $1.87 \times 10^{-4}$ | 48.7 |

[a]Conditions: in CH$_2$Cl$_2$, [c1-Val-OH]/[TBD] = 0.47 mM/0.57 mM, [c1-Val-OH]/[TBD]/[TFA] = 0.45 mM/0.54 mM/0.54 mM. [b]Molar ratios of [($P$)-c1-Val-X]/[($M$)-c1-Val-X] (X = OH or O$^-$) at the thermodynamic equilibrium at 5, 15, and 35 °C are assumed to be identical to that at 25 °C. [c]Molar ratio of [($P$)-c1-Val-X]/[($M$)-c1-Val-X] (X = OH or O$^-$) at the thermodynamic equilibrium at 25 °C determined by the CD intensity at 224 nm ($\Delta\varepsilon_{224}$) using the following equation: $\Delta\varepsilon_{224} = -\alpha|\Delta\varepsilon_{224max}| + (1-\alpha)|\Delta\varepsilon_{224max}|$, where $\alpha$ is mole fraction of ($P$)-c1-Val-OH. [d]Estimated from the time-dependent CD intensity changes (Supplementary Fig. 22). [e]Estimated from the equation $K = [(M)$-c1-Val-X]/[$(P)$–c1-Val-X] = $k_1/k_2$. [f]Half-life time ($t_{1/2}$) was obtained from the following equation: $t_{1/2}$ (min) = ln2/(($k_1 + k_2$) × 60) = 0.693/(($k_1 + k_2$) × 60).

**Table 2 | Thermodynamic activation parameters for the interconversion between singly and doubly stapled ($P$)- and ($M$)-α-helical peptides[a]**

| Peptide | $E_a$ (kJ mol$^{-1}$) | $\Delta G^{\ddagger}_{20}$ (kJ mol$^{-1}$) | $\Delta H^{\ddagger}$ (kJ mol$^{-1}$) | $\Delta S^{\ddagger}$ (J mol$^{-1}$ K$^{-1}$) |
|---|---|---|---|---|
| c1-Val-O$^-$ ($P$ to $M$) | $92.4 \pm 0.6$ | $85.3 \pm 1.2$ | $90.0 \pm 0.6$ | $15.6 \pm 2.1$ |
| c1-Val-O$^-$ ($M$ to $P$) | | $88.5 \pm 1.2$ | | $5.1 \pm 2.1$ |
| c1-Val-OH ($P$ to $M$) | $105 \pm 5$ | $90.5 \pm 10.6$ | $103 \pm 5$ | $42 \pm 18$ |
| c1-Val-OH ($M$ to $P$) | | $87.2 \pm 10.6$ | | $53 \pm 18$ |
| dc2-Val-OH ($P$ to $M$) | $133 \pm 7$ | $122 \pm 13$ | $130 \pm 7$ | $26 \pm 20$ |
| dc2-Val-OH ($M$ to $P$) | | $120 \pm 13$ | | $35 \pm 20$ |
| dc2-Aib-OMe | $131 \pm 3$ | $120 \pm 5$ | $128 \pm 3$ | $30.4 \pm 7.9$ |

[a]Estimated by Arrhenius and Eyring plots using the kinetic data in Tables 1, 3, and 4

### Acid/base-triggered reversible helicity inversion of singly stapled α-helical c1-Val-OH

As discussed above, **c1-Val-OH** adopted the α-helical conformation in CH$_2$Cl$_2$ with an excess of the ($P$)-handedness induced by the specific hydrogen bond of the C-terminal carboxy proton. We anticipated that its deprotonation by bases would induce a flip to the opposite ($M$)-helix because of the disfavored O···O interaction between the carboxylate oxygen atom and the carbonyl oxygen atom of the third or fourth amino acid residue from the C-terminus in a manner similar to that of **c1-Val-O$^t$Bu** (Fig. 1b). If the $P/M$ interconversion of **c1-Val-OH** is slow, the inversion kinetics upon the addition of bases could be monitored by CD measurements.

With this in mind, we performed time-dependent CD measurements of **c1-Val-OH** upon the addition of 1,5,7-triazabicyclo[4.4.0]dec-5-ene (TBD) as a strong organic base that can effectively deprotonate the C-terminal hydrogen-bonded carboxy proton. As expected, upon introducing TBD (1.2 equiv), the CD signal at 224 nm for **c1-Val-OH** in CH$_2$Cl$_2$ at 25 °C changed slowly from negative to positive, whereas no change was observed for the C-terminal protected **c1-Val-O$^t$Bu** (Fig. 2a, b (ii, v)). This suggested the base-induced helicity inversion of **c1-Val-OH** with the change in the $P/M$ molar ratio from 78/22 to 22/78 (see Methods for details). Moreover, the resulting CD spectrum of **c1-Val-OH** with TBD almost completely reverted back to the original CD spectrum upon the addition of 1.2 equiv of trifluoroacetic acid (TFA) at 25 °C (Fig. 2a, b (iii)). In these processes, the deprotonation of **c1-Val-OH** by TBD gave the **c1-Val-O$^-$·H$^+$-TBD** salt, which was readily protonated by TFA to form **c1-Val-OH**. The linear regression analyses of the logarithm of the CD intensities of these obtained CD data indicated that the $P/M$ interconversion obeyed the first-order kinetic model shown in Fig. 2 (Supplementary Fig. 22) and was much slower than the acid/base-triggered deprotonation and protonation reactions of **c1-Val-OH**.

The $P/M$ interconversion rates ($k_1$ (from $P$ to $M$) and $k_2$ (from $M$ to $P$) (sec$^{-1}$)) for the acid/base-triggered helicity inversion processes for **c1-Val-O$^-$** (with TBD) and **c1-Val-OH** (with a 1:1 mixture of TBD and TFA) at 25 °C are summarized in Table 1. Interestingly, $k_1$ and $k_2$ for **c1-Val-OH** are slower than those for **c1-Val-O$^-$** (Table 1). This rate difference is most likely because the strongly basic TBD not only deprotonates the C-terminal carboxy proton but also interacts with the acidic amide NH protons to destabilize the intramolecular hydrogen bonds in **c1-Val-OH**, as supported by the significant broadening of some of the amide NH proton signals in the $^1$H NMR spectrum of **c1-Val-OH** with 1.2 equiv of TBD in CD$_2$Cl$_2$ (Supplementary Fig. 24). Arrhenius and Eyring plots using the kinetic data from Table 1 provided the thermodynamic activation parameters (Supplementary Fig. 23), which are summarized in Table 2. The $\Delta S^{\ddagger}$ value for **c1-Val-O$^-$** is smaller than that for **c1-Val-OH** probably due to the aforementioned destabilization of the helical structure by TBD. The activation energy ($E_a$) for the $P/M$ interconversion of **c1-Val-OH** (105 kJ mol$^{-1}$) is higher than that of the reported dynamic stapled 3$_{10}$-helical peptide (97.7 kJ mol$^{-1}$) with the cross-linking between $i$ and $i+3$ positions by a rigid cross-linker, although the number of the amino acid residues within the stapled region ($i$, $i+7$) in the α-helical **c1-Val-OH** is twice that of the stapled 3$_{10}$-helical peptide[31]. Moreover, the rate of the $P/M$ interconversion of **c1-Val-OH** is approximately 10$^6$ times slower than that of the previously reported non-stapled 3$_{10}$-helical octapeptide Fmoc-(Aib)$_8$-O$^t$Bu (Fmoc = 9-fluorenylmethyloxycarbonyl group)[30].

As described above, the preferred helix sense of **c1-Val-OH** can be switched from $P$ to $M$ and vice versa by changing the solvent from TFE or CH$_2$Cl$_2$ to MeOH. Thus, the kinetic and thermodynamic activation parameters in a protic solvent were obtained by monitoring the CD intensity changes upon dilution with a different kind of solvent. The activation energy ($E_a$) value of **c1-Val-OH** in MeOH/TFE (49/1, v/v) was estimated to be 89.7 kJ mol$^{-1}$, which is -15 kJ mol$^{-1}$ lower than that in

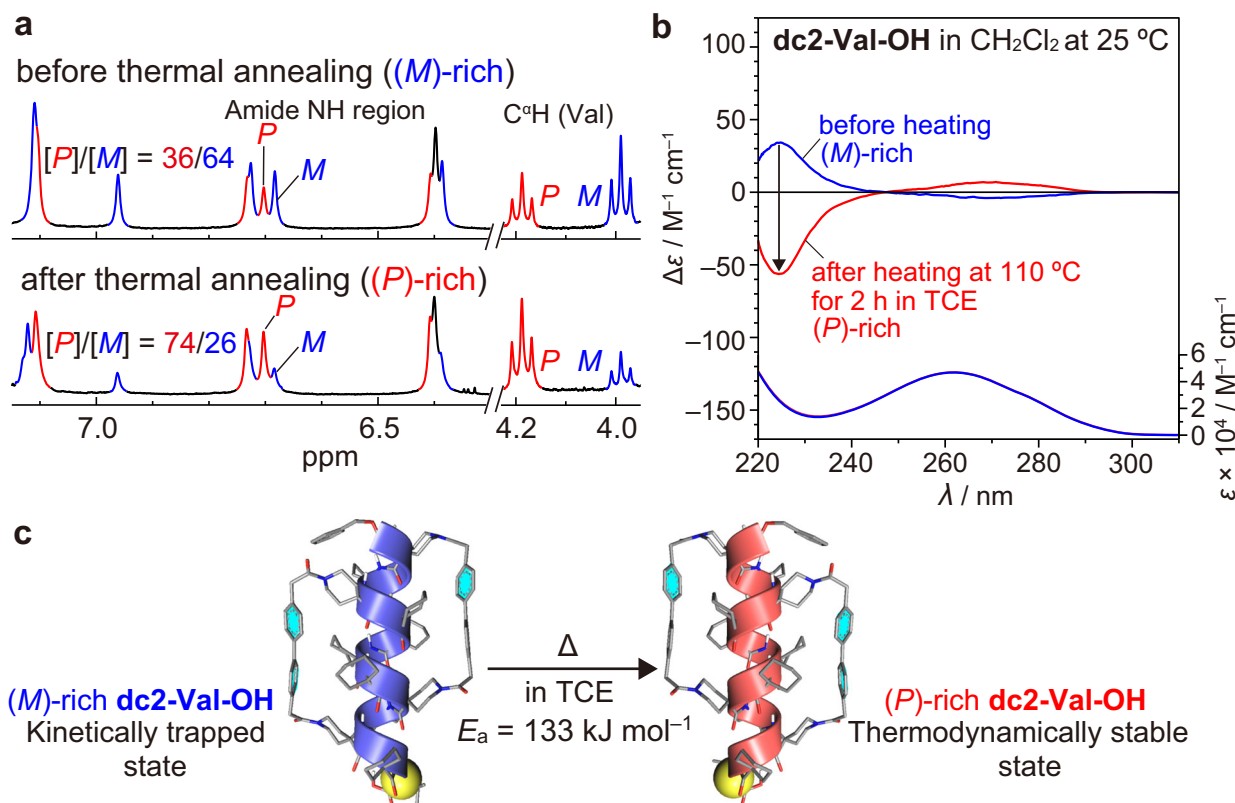

**Fig. 4 | Thermal annealing of dc2-Val-OH. a** ¹H NMR (400 MHz, 3.0 mM) spectra of the as-obtained **dc2-Val-OH** (top) and the annealed **dc2-Val-OH** (bottom) in CD₂Cl₂/(CD₃)₂SO (19/1, v/v) measured at 308 K. The annealed **dc2-Val-OH** was prepared by heating the as-obtained **dc2-Val-OH** in freshly distilled 1,1,2,2-tetrachloroethane (TCE) at 110 °C for 2 h followed by removal of the solvent under reduced pressure. For detailed experimental procedures, see Methods. **b** CD (top)

and absorption (bottom) spectra of the as-obtained **dc2-Val-OH** (blue line) and the annealed **dc2-Val-OH** (red line) in CH₂Cl₂ at 25 °C: [**dc2-Val-OH**] = 0.22–0.26 mM. The CD spectral patterns of the as-obtained **dc2-Val-OH** in TFE and MeOH are similar to those of **c1-Val-OᵗBu** and **c1-Val-OH** (Supplementary Fig. 32). **c** Schematic representation of the transformation of the kinetically trapped (as-obtained) (*M*)-rich **dc2-Val-OH** into the thermodynamically stable (annealed) (*P*)-rich **dc2-Val-OH**.

CH₂Cl₂ (Supplementary Fig. 25b–d and Supplementary Table 12). A similar protic solvent-induced acceleration of the *P/M* interconversion is also known[53].

Other strong organic bases, such as 1,8-diazabicyclo[5.4.0]undec-7-ene (DBU) (1.2 equiv) and the amidine **A3** (2.2 equiv), also induced an excess of the (*M*)-handedness in **c1-Val-OH** with the *P/M* molar ratios of 8/92 and 18/82, respectively, whereas the *P/M* molar ratio of **c1-Val-OH** almost remained unchanged upon the addition of 1.2 equiv of aliphatic primary, secondary and tertiary amines (ⁿBuNH₂, NHEt₂, and NEt₃, respectively) (Fig. 2c and Supplementary Fig. 28). This negligible effect is most likely due to the low association constant (*K*ₐ, M⁻¹) of **c1-Val-OH** with these aliphatic amines (*K*ₐ ≤ 10²). In contrast, the *K*ₐ values of **c1-Val-OH** with TBD, DBU, and **A3** were determined to be >10³ (for determination of the *K*ₐ values, see the Supplementary Information (section 11)).

**Transformation of kinetically trapped helix sense of doubly stapled α-helical dc2-Val-OH into the thermodynamically stable one**

Based on the above results, the doubly stapled **dc2-Val-OH** was expected to show very slow *P/M* interconversion. In this case, the *P/M* molar ratio of the as-obtained **dc2-Val-OH**, which was synthesized by the double stapling of an optically active dodecapeptide with the biphenyl-based cross-linking reagent, would not necessarily reach a plateau value. Thus, this ratio would be different from that of the singly stapled analog **c1-Val-OH** at the thermodynamic equilibrium state. Interestingly, the CD and ¹H NMR measurements revealed that the as-obtained **dc2-Val-OH** adopted an excess of the (*M*)-α-helical conformation with the *P/M* molar ratio of 36/64 (Fig. 4a, b), whose excess

handedness was opposite to that of **c1-Val-OH**. This suggested that the (*M*)-rich **dc2-Val-OH** was kinetically trapped during the cross-linking reaction of the corresponding non-stapled peptide with an excess of an (*M*)-helical conformation, which appeared to be induced by an intramolecular acid-base interaction between the side-chain piperidine and C-terminal carboxy groups. In fact, the thermal annealing of the as-obtained **dc2-Val-OH** in 1,1,2,2-tetrachloroethane (TCE) at 110 °C resulted in an inversion of the excess handedness from the (*M*)-helix to the (*P*)-helix, with the *P/M* molar ratio changing from 36/64 to 74/26 (Fig. 4) (for details of the experimental procedures, see Methods), indicating the conversion from the kinetically trapped state to the thermodynamically stable state. Thus, the time-dependent changes in the CD intensity of the as-obtained **dc2-Val-OH** at 80, 90, 100, 110 °C were monitored, and the subsequent Arrhenius and Eyring analyses using the obtained kinetic data provided the thermodynamic activation data (Tables 2 and 3 and Supplementary Figs. 30 and 31). The obtained Δ*H*‡ value for the *P/M* interconversion of **dc2-Val-OH** is much higher than that of **c1-Val-OH**, but both Δ*S*‡ values are positive and not much different from each other, implying that the helix inversion of the singly and doubly stapled peptides took place through the formation of a distorted intermediate structure by breaking of multiple intramolecular hydrogen bonds (Table 2). Surprisingly, the rate constant (*k₁* + *k₂*) for the interconversion between the (*P*)- and (*M*)-α-helices of **dc2-Val-OH** at 25 °C was estimated to be approximately 10⁶ times slower than that of the singly stapled analog, **c1-Val-OH**, by extrapolation using the Arrhenius-type equation (Tables 1 and 3). The corresponding *t*₁/₂ value for **dc2-Val-OH** was 1.33 × 10⁶ min (ca. 920 days), clearly indicating that the double stapling strategy can transform the dynamic α-helical peptides into (quasi-)static ones. It

**Table 3 | Rate constants ($k_1$ and $k_2$, sec$^{-1}$) and half-life time ($t_{1/2}$, min) for the interconversion between (P)- and (M)-dc2-Val-OH[a]**

| Temp (°C) | [P]/[M] | $k_1 + k_2$ (sec$^{-1}$)[c] | $k_1$ (sec$^{-1}$)[e] | $k_2$ (sec$^{-1}$)[e] | $t_{1/2}$ (min)[f] |
|---|---|---|---|---|---|
| 110 | 74/26[b] | $1.17 \times 10^{-3}$ | $3.10 \times 10^{-4}$ | $8.63 \times 10^{-4}$ | 9.8 |
| 100 | 74/26[c] | $5.03 \times 10^{-4}$ | $1.33 \times 10^{-4}$ | $3.70 \times 10^{-4}$ | 23.0 |
| 90 | 74/26[c] | $1.23 \times 10^{-4}$ | $3.25 \times 10^{-5}$ | $9.03 \times 10^{-5}$ | 94.1 |
| 80 | 74/26[c] | $3.65 \times 10^{-5}$ | $9.66 \times 10^{-6}$ | $2.69 \times 10^{-5}$ | 316 |
| 25 | 74/26[c] | $8.67 \times 10^{-9\,d}$ | $2.29 \times 10^{-9\,d}$ | $6.38 \times 10^{-9\,d}$ | $1.33 \times 10^6$ |

[a]Conditions: in 1,1,2,2-tetrachloroethane, [**dc2-Val-OH**] = 0.51 mM. [b]Molar ratio of [(P)-**dc2-Val-OH**]/[(M)-**dc2-Val-OH**] at the thermodynamic equilibrium at 110 °C determined using $^1$H NMR spectroscopy (Fig. 4a). [c]Molar ratios of [(P)-**dc2-Val-OH**]/[(M)-**dc2-Val-OH**] at the thermodynamic equilibrium within the temperature range between 25–110 °C are assumed to be constant. [d]Extrapolated from the following Arrhenius-type equations: $k_1 = \exp(-16000/T + 33.79)$ and $k_2 = \exp(-16000/T + 34.81)$ (Supplementary Fig. 31a), where $T$ is the absolute temperature. [e]Estimated from the following equation: $K = [(M)\text{-}\textbf{dc2-Val-OH}]/[(P)\text{-}\textbf{dc2-Val-OH}] = k_1/k_2$. [f]Half-life time ($t_{1/2}$) was obtained from the following equation: $t_{1/2}$ (min) = $\ln 2/((k_1+k_2) \times 60) = 0.693/((k_1+k_2) \times 60)$.

should be noted that a protic solvent did not significantly accelerate the P/M interconversion of **dc2-Val-OH** ($E_a = 129$ kJ mol$^{-1}$), demonstrating the effectiveness of the double stapling strategy for constructing static α-helical peptides (Supplementary Fig. 33 and Supplementary Table 13).

## Optical resolution of rac-dc2-Aib-OMe and its racemization kinetics

As discussed above, the doubly stapled α-helical peptides display very slow P/M interconversion. This time scale is sufficiently slow that the P/M interconversion would be negligible during chromatographic separation at room temperature. Thus, it is possible to prepare one-handed static α-helical peptides without the aid of any chiral auxiliary simply by optical resolution of the corresponding racemic peptides. To this end, we performed the optical resolution of rac-**dc2-Aib-OMe** consisting only of achiral components by HPLC on a chiral stationary phase to obtain enantiopure (P)- and (M)-α-helical **dc2-Aib-OMe** (Fig. 1a). The HPLC analyses indicated the complete separation of the racemate into the (P)- and (M)-**dc2-Aib-OMe** enantiomers; their optical purities were almost ±100% enantiomeric excess (e.e. (%) = ([P] − [M])/([P] + [M]) × 100) (Fig. 5a). This was also confirmed by the CD measurements of the enantiomers, where their CD spectra were exact mirror images of each other (Fig. 5b). The kinetic (the rate constant for the racemization ($k_{rac}$, sec$^{-1}$) instead of $k_1$ and $k_2$ ($2k_{rac} = k_1 + k_2$)) and thermodynamic activation parameters were obtained and are summarized in Tables 2 and 4 (Supplementary Figs. 34 and 35). The P/M interconversion rate of **dc2-Aib-OMe** is similar to that of **dc2-Val-OH**, suggesting that the C-terminal group has only a small effect on the interconversion rate. Nevertheless, the $E_a$ value for the P/M interconversion (racemization) of the one-handed **dc2-Aib-OMe** is slightly lower than that of **dc2-Val-OH**, most likely due to the absence of the C-terminal carboxy proton which participates in the intramolecular hydrogen bond. The estimated $k_{rac}$ value of the one-handed **dc2-Aib-OMe** at 25 °C is ~$10^{12}$ times smaller than that of the previously reported non-stapled homo-Aib octapeptide[30] and implies that it would take ~8.7 years ($t_{1/2}$ = ca. 1.5 years) to lose 98% of the optical activity (e.e.) at 25 °C (Fig. 5c).

## Discussion

In conclusion, a double stapling strategy was effectively used to transform dynamic helical peptides into (quasi-)static ones. The double stapling of the achiral peptide chains at appropriate positions induces the well-defined α-helical conformation together with a drastic increase of the energy barrier for the P/M helix interconversion, which takes place only on a time scale of years at room temperature. Owing to this (quasi-)static feature, enantiopure one-handed α-helical peptides composed only of achiral peptides and rigid cross-linkers were successfully obtained by optical resolution of the corresponding racemate. These one-handed peptides showed optical activity originating only from the α-helical conformation of the achiral main chain. The double stapling of the corresponding achiral peptide with the unprotected L-Val-OH residue at the C-terminus was found to

produce the doubly stapled α-helical peptide with an excess (M)-handedness as a kinetically trapped state. This was successfully transformed into a thermodynamically stable (P)-rich α-helical peptide with a very high energy barrier for the P/M helix interconversion in both protic and aprotic solvents by heating at high temperatures. On the other hand, the corresponding singly stapled peptide exhibits interconversion between the (P)- and (M)-α-helices on a time scale of minutes, and its preferred helix sense can be reversibly switched by deprotonation/protonation of the C-terminal carboxy group in response to a suitable base or acid, or by changing solvents. These simple helicity inversion systems in combination with the stapling strategy described in this paper can control both the dynamics and helix sense of the α-helical peptides. In addition, these well-defined α-helical scaffolds, whose helical handedness is sensitive to chiral stimuli, may facilitate our deeper understanding of the role of chirality in the α-helix. These findings may contribute to the development of α-helical peptide-based drugs[33,45,46], asymmetric catalysts[54,55], and next-generation chiral materials[56], which may require robust α-helical structures.

## Methods

### Kinetic and thermodynamic analyses of acid/base-triggered reversible helicity inversion of c1-Val-OH

Stock solutions of **c1-Val-OH** (0.50 mM) (solution **I**), TBD (10.0 mM) (solution **II**), other organic bases (solution **II'**), and TFA (10.0 mM) (solution **III**) were prepared in CH$_2$Cl$_2$. A 24 μL aliquot of **II** or **II'** (1.2 equiv to **c1-Val-OH**) was added to a 0.1 cm quartz cell containing a 400 μL aliquot of **I** at 5, 15, 25, or 35 °C, and time-dependent CD intensity changes were monitored starting immediately after mixing. After reaching an equilibrium, to this (total 424 μL) was added a 24 μL aliquot of **III** (1.2 equiv to **c1-Val-OH**), and time-dependent CD intensity changes were then monitored starting immediately after mixing.

The observed time-dependent CD intensity changes indicated that the P/M interconversion obeyed the first-order kinetic model shown in Fig. 2, in which $k_1$ (from P to M) and $k_2$ (from M to P) (sec$^{-1}$) are the rate constants for the interconversion between diastereomeric (P)- and (M)-**c1-Val-OH** (in the absence and presence of a 1:1 mixture of TBD and TFA) or between diastereomeric (P)- and (M)-**c1-Val-O**$^-$ (in the presence of TBD). The linear regression analysis of the logarithm of the CD intensities of the obtained CD data gave the total rate constants ($k_1 + k_2$) (Supplementary Fig. 22 and Table 1). The half-life time ($t_{1/2}$, min) was obtained from Eq. (1):

$$t_{1/2} = \ln 2/((k_1 + k_2) \times 60) = 0.693/((k_1 + k_2) \times 60) \tag{1}$$

Each value of $k_1$ and $k_2$ was estimated from Eq. (2):

$$K = [M]/[P] = k_1/k_2 \tag{2}$$

where $K$ is M/P equilibrium constant.

The $K$ value was estimated by the following procedures: the CD intensity at 224 nm ($\Delta\varepsilon_{224}$), which reflects the helix sense excess after

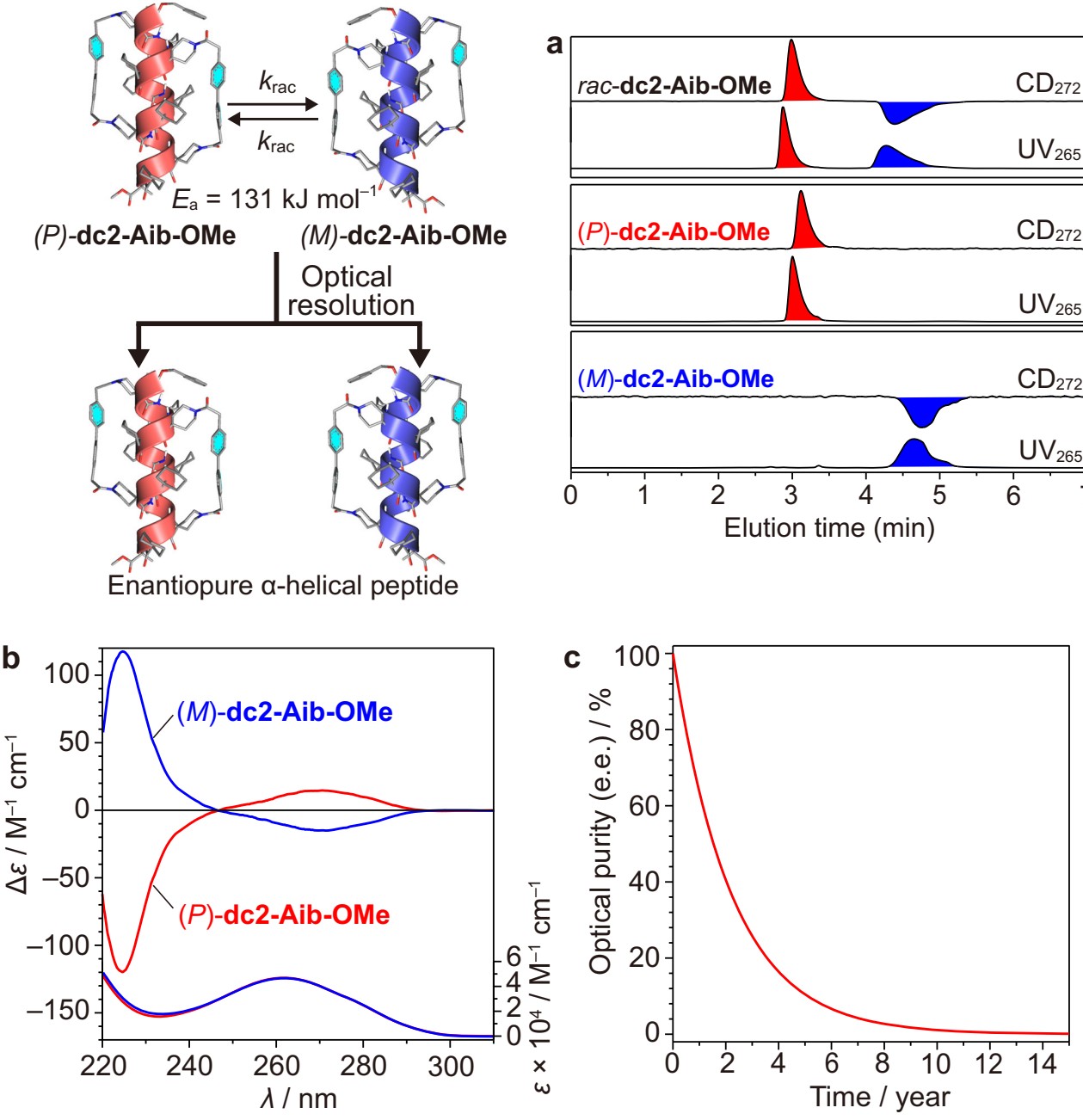

**Fig. 5 | Optical resolution of *rac*-dc2-Aib-OMe. a** CD ($\lambda$ = 272 nm) and UV ($\lambda$ = 265 nm) detected HPLC chromatograms of *rac*-**dc2-Aib-OMe** (top), (*P*)-**dc2-Aib-OMe** (middle) and (*M*)-**dc2-Aib-OMe** (bottom). HPLC conditions: column, CHIRALPAK IB N-5 (DAICEL, 0.46 (i.d.) × 25 cm); eluent, CH$_2$Cl$_2$/$^i$PrOH (97/3, v/v); flow rate, 1.0 mL/min; column temperature, 25 °C. **b** CD (top) and absorption (bottom) spectra of enantiopure (*P*)- and (*M*)-**dc2-Aib-OMe** in CH$_2$Cl$_2$ at 25 °C: [**dc2-Aib-OMe**] = 0.28 mM. **c** Simulated kinetic curve for the racemization of enantiopure **dc2-Aib-OMe** at 25 °C using the equation e.e. = 100·exp(−2$k_{rac}$t·(60·60·24·365)).

reaching equilibrium, is the sum of the CD intensities of the two diastereomers (*P*)- and (*M*)-**c1-Val-X**. The CD spectral patterns of (*P*)-rich **c1-Val-OH** and (*M*)-rich **c1-Val-O$^t$Bu** are almost mirror images of each other (Supplementary Fig. 21a). Therefore, their absolute maximum CD intensities at 224 nm (|$\Delta\varepsilon_{224max}$|) are assumed to be equal and then,

$$\Delta\varepsilon_{224} = -\alpha|\Delta\varepsilon_{224\,max}| + \beta|\Delta\varepsilon_{224\,max}| \qquad (3)$$

$$\alpha + \beta = 1 \qquad (4)$$

where $\alpha$ and $\beta$ are mole fractions of (*P*)- and (*M*)-**c1-Val-X**.

The mole fractions of (*P*)- and (*M*)-**c1-Val-OH** and (*P*)- and (*M*)-**c1-Val-O$^t$Bu** in CD$_2$Cl$_2$ at 298 K were estimated to be 0.78 and 0.22 and 0.38 and 0.62, respectively, from the integral ratios of these diastereomeric pairs in the $^1$H NMR spectra (Supplementary Fig. 21b, c). In addition, $\Delta\varepsilon_{224}$ values of **c1-Val-OH** and **c1-Val-O$^t$Bu** in CH$_2$Cl$_2$ at 25 °C were −43.8 and +18.8 M$^{-1}$ cm$^{-1}$, respectively (Supplementary Fig. 21a). Using these values and Eqs. (3) and (4), the $\Delta\varepsilon_{224max}$ values of **c1-Val-OH** and **c1-Val-O$^t$Bu** were estimated to be −78.2 and +78.3 M$^{-1}$ cm$^{-1}$, respectively, and thus their absolute values are almost equal. Therefore, the molar ratios of [(*M*)-**c1-Val-X** (X = **OH** or **O$^-$**)]/[(*P*)-**c1-Val-X**] (= *M*/*P* equilibrium constant (*K*)) at 25 °C were estimated from the CD intensity at 224 nm ($\Delta\varepsilon_{224}$) using Eqs. (3) and (4) and the |$\Delta\varepsilon_{224max}$|

**Table 4 | Rate constants ($k_{rac}$, sec$^{-1}$) and half-life time ($t_{1/2}$, min) for the racemization of (*M*)-dc2-Aib-OMe[a]**

| temp (°C) | $k_{rac}$ (sec$^{-1}$) | $t_{1/2}$ (min)[d] |
|---|---|---|
| 100 | $3.23 \times 10^{-4}$ [b] | 17.9 |
| 90 | $9.05 \times 10^{-5}$ [b] | 63.8 |
| 80 | $2.72 \times 10^{-5}$ [b] | 213 |
| 70 | $7.81 \times 10^{-6}$ [b] | 739 |
| 25 | $7.16 \times 10^{-9}$ [c] | $8.07 \times 10^{5}$ |

[a]Conditions: in 1,1,2,2-tetrachloroethane, [**dc2-Aib-OMe**] = 0.46 mM. [b]Estimated from the time-dependent CD intensity changes (Supplementary Fig. 34). [c]Extrapolated from the following Arrhenius-type equation: $k_{rac}$ = exp(–15810/$T$ + 34.30) (Supplementary Fig. 35a). [d]Half-life time ($t_{1/2}$) was obtained from the following equation: $t_{1/2}$ (min) = ln2/2($k_{rac} \times 60$) = 0.693/2($k_{rac} \times 60$).

value of 78. Because the mole fractions of (*P*)- and (*M*)-**c1-Val-OH** were almost constant in the temperature range between 278 and 308 K (Supplementary Fig. 21b), the molar ratios of (*P*)- and (*M*)-**c1-Val-OX** (**X** = **OH** or **O$^-$**) at 5, 15, 35 °C were assumed to be identical to that at 25 °C.

The obtained $k_1$ and $k_2$ values were analyzed according to Arrhenius Eq. (5):

$$\ln k = \ln A - E_a/RT \qquad (5)$$

where $A$ (sec$^{-1}$), $E_a$ (kJ mol$^{-1}$), $R$ ($8.314 \times 10^{-3}$ kJ K$^{-1}$ mol$^{-1}$), and $T$ (K) are the preexponential factor, the activation energy, the gas constant, and the absolute temperature, respectively.

### Kinetic and thermodynamic analyses of the transformation of the kinetically trapped (*M*)-rich dc2-Val-OH into the thermodynamically stable (*P*)-rich dc2-Val-OH

A solution of the kinetically trapped (*M*)-rich **dc2-Val-OH** (0.51 mM) in freshly distilled TCE was heated to 80, 90, 100, and 110 °C, and the CD intensity changes at 272 nm ($\theta_{272}$) were monitored at each temperature after reaching the desired temperature (Supplementary Fig. 30a). In these measurements, the CD intensity changes were monitored at 272 nm instead of 224 nm, because the strong absorption of TCE, used as a high-boiling solvent, prevents CD measurements below 250 nm.

The observed time-dependent CD intensity changes indicated that the *P/M* interconversion obeyed the first-order kinetic model shown in Supplementary Fig. 29, where $k_1$ (from *P* to *M*) and $k_2$ (from *M* to *P*) (sec$^{-1}$) are the rate constants for the interconversion between diastereomeric (*P*)- and (*M*)-**dc2-Val-OH**. Linear regression analysis of the logarithm of the CD intensities of the obtained CD data gave the total rate constants ($k_1 + k_2$) (Supplementary Fig. 30b and Table 3). The half-life time ($t_{1/2}$, min) and each value of $k_1$ and $k_2$ were obtained from Eqs. (1) and (2), respectively. The $K$ value was estimated by the following procedures: the above solution was heated at 110 °C for 2 h to reach an equilibrium between (*P*)- and (*M*)-**dc2-Val-OH**, as confirmed by the time-dependent CD measurement (Supplementary Fig. 30a). The solvent was then evaporated to dryness under reduced pressure. The residue was dissolved in CD$_2$Cl$_2$/(CD$_3$)$_2$SO (19/1, v/v), and the $^1$H NMR measurement was performed immediately after the dissolution, during which a change in the molar ratio was negligible (Fig. 4a). Similar to the case of **c1-Val-OX** (**X** = **OH** or **O$^-$**), the molar ratios of (*P*)- and (*M*)-**dc2-Val-OH** at 80, 90, and 100 °C were assumed to be identical to that at 110 °C. The obtained $k_1$ and $k_2$ values were analyzed according to the Arrhenius Eq. (5).

### Kinetic and thermodynamic analyses of the racemization of (*M*)-dc2-Aib-OMe

A solution of enantiopure (*M*)-**dc2-Aib-OMe**, obtained through optical resolution by using HPLC on a chiral stationary phase (a semi-preparative CHIRALPAK IB N-5 column (DAICEL, 1.0 (i.d.) × 25 cm); eluent, CH$_2$Cl$_2$/$^i$PrOH (97/3, v/v); flow rate, 3.0 mL/min; column

temperature, 25 °C), in freshly distilled 1,1,2,2-tetrachloroethane was heated to 100 °C, and the CD intensity change at 272 nm ($\theta_{272}$) was monitored after reaching the desired temperature. The temperature of this solution of the partially racemized **dc2-Aib-OMe** was then changed to 90, 80, 70 °C in a stepwise manner, and the $\theta_{272}$ values at each temperature were monitored after reaching the desired temperature (Supplementary Fig. 34a). The CD intensity changes were monitored at 272 nm instead of 224 nm, because the use of TCE as a high-boiling solvent prevents the CD measurement below 250 nm.

The obtained time-dependent CD intensity changes indicated that the *P/M* interconversion obeyed the first-order kinetic model shown in Fig. 5, where $k_{rac}$ (sec$^{-1}$) is the rate constant for the racemization. Linear regression analysis of the logarithm of the CD intensities of the obtained CD data gave the rate constants ($k_{rac}$) (Supplementary Fig. 34b and Table 4). The half-life time ($t_{1/2}$, min) was obtained from Eq. (6):

$$t_{1/2} = \ln 2/(2k_{rac} \times 60) = 0.693/(2k_{rac} \times 60) \qquad (6)$$

## Data availability

The data supporting the results of this study are available in the paper and Supplementary files, and from the corresponding authors upon request. Source data underlying Fig. 2b and Supplementary Figs 16b, 20b, 22, 23, 25b–d, 26a, d, 30, 31, 33b–d, 34, and 35 and Cartesian coordinates for all calculated structures are provided as a Source Data file. The X-ray crystallographic data for the structure reported in this study has been deposited at the Cambridge Crystallographic Data Center (CCDC) under deposition number 2261608. These data can be obtained free of charge from The Cambridge Crystallographic Data Center via http://www.ccdc.cam.ac.uk/data_request/cif. Source data are provided in this paper.

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

## Acknowledgements

This work was supported in part by JSPS KAKENHI (JP22K05220 to NO and JP21H05477 and JP23H04021 to SA [Condensed Conjugation]) and the World Premier International Research Initiative, Ministry of Education, Culture, Sports, Science and Technology, Japan (NO, MJM, and SA). The authors thank professor Katsuhiro Maeda (Kanazawa University) for the use of his HPLC system and chiral columns.

## Author contributions

N.O. conceived the project, designed and performed the experiments, and analyzed the data. S.A. performed X-ray crystallographic analysis. All authors co-wrote the paper.

## Competing interests

The authors declare no competing interests.

## Additional information

**Peer review information** : *Nature Communications* thanks Jonathan Clayden, and the other, anonymous, reviewers for their contribution to the peer review of this work. A peer review file is available.

