## [Peer Review File · Nature Communications]

Stapling strategy for slowing helicity interconversion of α -helical peptides and isolating chiral auxiliary-free one-handed formsREVIEWER COMMENTS

Reviewer #1 (Remarks to the Author):

This paper concerns the synthesis of alpha-helical peptides containing no chiral amino acids and their resolution into single screw sense conformers.

All-achiral 310 helical and alpha helical peptides are known. The barrier between alpha and 310 structures are very low, as so little structural reorganisation is required to move from one to the other. Small changes in solvent mixtures can lead to the switch, and crystallography is no guide to solution structure. Recent evidence has shown that aggregation can control the alpha/310 helical preference too (Woolfson, Nature 2022, 607, 387 – this could be cited)

Thermodynamically induced helicity is well established, but the resulting diastereoisomeric pairs typically interconvert rapidly; removal of the asymmetry inducing the helix leads to racemisation. However, Yashima, Maeda and others have reported several examples where this is not the case: these are not peptides, but nonetheless deserve citation – especially those polymers that can be induced into a single screw sense by a chiral guest and their stabilities have a very significant dependence on solvent Eg doi/10.1002/anie.202217020

Helix stapling is well established to lower the rate of interconversion of helical peptides, as reported by one of the current authors in 2008. In that instance, the stapling linked residue is separated by 3 units, consistent with the formation of a 310 helix. The racemisation half life was of the order of 1 minute at room temperature.

In the current paper, stapling is used to slow helix into conversion in similar structures, but the staple spans 7 residues, consistent with adoption of an alpha-helical structure. The resulting helices interconvert extremely slowly.

On p9, the CD of the single stapled peptide is referred to. Alpha and 310 helices may be characterised by the relative intensity of the bands at ~ 208 into 210 nm. For 310 helices, the band at 220 nm can vary in sign between positive and negative (compare J. Am. Chem. Soc. 1996, 118, 11, 2744–2745 with Nature 2022, 607, 387 and discussion in E. Longo , A. Moretto , F. Formaggio and C. Toniolo , Chirality, 2011, 23 , 756 –760 and footnote 30 of Chem. Commun., 2020, 56, 12049-12052) , for reasons that are still not yet clear, but nonetheless the band at 208 nm is diagnostic of screw sense. Because they have been determined in chloroform, but the CDs in this paper do not show wavelength below 220 nm, which makes the detailed discussion of conformation in solution using CD impossible (changes in screw sense inversion of elicited is detectable, but not absolute screw sense for conformational preference). It also means that the comparison between the CDs of singly and doubly stapled peptides is not simple matter of screw sense reversal – it is possible that subtle differences between alpha and 310 preferences are at play here as well. The CD should be repeated in solvents that allow shorter wavelength determination of molar ellipticity. The band at 260 nm is presumably due to the biphenyl staple. This deserves comment.

Barrier determinations also appear to have been done in CH_2Cl_2 , which is not at all biomimetic, although it can be used as a model for phospholipid membranes. Similar determination should be carried out for protic solvents, especially given the suggestion from the authors that these molecules have the potential to be used in the context of medicinal chemistry.

Likewise, the NMR experiments in indicate alpha-helical structures, but given the solvent dependence of 310 and alpha-helical preferences it should be explored whether this is consistent across other more protic solvents.

Overall these are interesting results, and the degree to which stapling slows the rate of helical inversion is significant, though perhaps unsurprising given the degree to which the peptide is rigidified by two staples. However there are many questions over the assignment of conformation

that need clarifying before the work becomes publishable.

Reviewer #2 (Remarks to the Author):

This is a very interesting contribution in the field of biotic foldamers done mainly by Naoki Ousaka who has a large experience in the field. Although the title suggest that the manuscript will be focused on achiral stapled foldamers, there is also a large part of the article, at least 70%, dealing with conformational communication along the foldamer chain that is derivatized at the C terminus with a chiral val residue. The authors clearly show how the pH affects to the conformational composition at the C-terminus chiral val residue, and how this changes affect to the screw sense excess of the foldamer. From these studies, they clearly show that while the single stapled foldamer is still dynamic, the double stapled one is quasi-static. There is no info about this part of the work neither in the abstract or title and can be missed during literature search.

The last part of the work deals with the isolation of both P and M helical structures from a completely achiral foldamer due to the double stapling strategy, which practically freeze the two P and M macromolecular helical conformers, making possible to isolate them and keep it stable in solution for a while.

I think that the research is very interesting for the scientific community working in biological chemistry, materials science, organic and inorganic chemistry (foldamers, metallofoldamers, peptides, DNA, helical covalent and supramolecular polymers). The work has been done carefully, all the data is clearly shown, and discussion is easy to follow.

Reviewer #3 (Remarks to the Author):

The manuscript by N. Ousaka and coworkers deals with the solution synthesis of six peptides, of which two made only of achiral residues. Most of the text refers to the Val-containing compounds termed c1-Val and dc2Val, however, therefore the title, abstract and conclusions do not reflect the results.

The description of a single-handed alpha-helix made of only achiral residues not obtained by any kind of chiral interactions is novel and deserves publication, but unfortunately this is not the case. There are several flaws in the present version of the manuscript:

1) Data on the peptides c1-Val-OH/OtBu and dc2-Val-OH/OtBu are not pertinent to the present article and should be removed totally. There are several studies in the literature on peptides containing just one chiral residue that adopt a single-handed helix. By the way, there are no clear data in support of those Val-containing peptides adopting a alpha-helical conformation, since CD has been performed in dichloromethane and the two maxima at 208 and 195 nm are not detectable. Both their position and relative intensity (especially $[\theta_{222}]/[\theta_{208}]$ ratio) could have given indications on the helix being alpha or 3-10. The authors claim (p.12, line 8) that the presence of all detectable sequential NH-NH correlations in the NOESY spectrum is diagnostic of a alpha-helix, but this observation is compatible with both kind of helices (see Wüthrich's book). By the way, apparently the authors did not produce the build-up curve to determine the mixing time for their NOESY, therefore there might be spin-diffusion, especially since the peptide concentration is quite high.

2) All studies on the Val-containing peptides presented here should have been performed on the Aib-containing ones instead. All paragraphs: "Conformational analysis of singly and doubly stapled peptides" (p.9) - M/P ratio determination by 1H-NMR included; "Acid/base-triggered reversible helicity inversion" (p.15) and related kinetics and thermodynamic parameters should be rewritten with data collected on the achiral peptide(s) dc2-Aib-OH/OMe.

There are also other, important issues to be tackled:

i) Figure 2, p.11 and par. "Acid/base-triggered reversible helicity inversion", p.15. There must be indications on the apparent pH of the solutions used. The differences between bulky, CHIRAL

(except for A3), cyclic amines and linear, achiral ones seem to imply that it is not really the basic pH that matters, but rather the association with the counterion, which acts as an 'auxiliary', with chirality on the nitrogen. "helicity inversion triggered by association with a counterion" seems a better choice for this phenomenon.

ii) Figure 2a: there's a band centered at about 270 nm, which is used afterwards to make several assumptions. Such a dichroic signal for helical peptides is usually due to the presence of aromatic groups experiencing environmental chirality (from the helix, other chiral centers nearby, etc). This means: 1. also the band at about 224nm can have a contribution by aromatic groups, therefore its intensity cannot be safely attributed to the conformation alone. 2. Since Val-OtBu does not show the same band, but the peptide backbone should be the same (Fig. 1c) then something is missing: either the presence of the additional H-bond c1-Val-OH strengthen the helix to the point that the aromatic groups becomes optically active and this does not happen for c1-Val-OtBu (although that 'aromatic' band is present also for dc2-Aib-OMe!), or all c1-Val-OH spectra (and dc2-Aib-OMe) were acquired in the presence of an aromatic contaminant able to closely interact with the peptide itself and not present in the solutions of c1-Val-OtBu. An explanation on this point is crucial, since - as already pointed out - the same band is present also for dc2-Aib-OMe. Moreover, the reasons why such a band has been chosen for the following studies are not clear and should be explained.

iii) Paragraph "Optical resolution of rac-dc2-Aib-OMe and its racemization kinetic", p.24-25. This is the core of the article, but the kinetics and thermodynamic parameters are not reported in the main article, and how they had been obtained not clearly explained. Once all the Val-peptides are removed from the article, the corresponding data for Aib-peptides should be added. The NMR (?) studies from which the M/P ratio and interconversion rate were extrapolated are not explained nor reported, but should be carried out if not yet done. Fig 24, p. S61 and Fig. 25, p. S62, refer to CD studies on the aromatic band at about 270nm and should be supported with the corresponding NMR study. Fig 24 at p. S61 is not sound, since the timeframe of the time-dependence CD is different among the temperatures. In particular, 10 minutes at 100°C is too short.

iv) the reasons why dichloromethane is used for the CD study must be included. Why not methanol or other solvents which are more transparent and show all the CD bands?

Some minor points:

-p.6, Figure 1c,d: R, R' at the C-terminus would avoid confusion. A table with the sequences would be helpful.

-final purity of the peptides should be provided somewhere.

- p. S14, lines 6-8. Washings with a CHCl₃/MeOH/EtOAc mixture is a non-standard procedure in solution-phase peptide synthesis, perhaps it should be further explained to allow for reproducibility.

- p. S28, supplementary figure 2. The complete (50-2500Da) mass spectra should be provided.

- p. S53, suppl. figure 18. The authors should state how they performed the M/P assignments. The 2D NMR spectrum of the same peptide c1-Val-OH, reported at p. S41, Figure 8 was acquired in another solvent and could not have been of help.

Response to Reviewer #1:

Thank you very much for your review and valuable comments for our manuscript. According to the suggestions, we have thoroughly revised our manuscript as follows. The changes are also indicated in the revised manuscript and Supplementary Information (SI) with a red color text.

Comment 1: This paper concerns the synthesis of alpha-helical peptides containing no chiral amino acids and their resolution into single screw sense conformers.

Reply 1: We agree with this comment.

Comment 2: All-achiral 3₁₀ helical and alpha helical peptides are known. The barrier between alpha and 3₁₀ structures are very low, as so little structural reorganisation is required to move from one to the other. Small changes in solvent mixtures can lead to the switch, and crystallography is no guide to solution structure. Recent evidence has shown that aggregation can control the alpha/3₁₀ helical preference too (Woolfson, Nature 2022, 607, 387 – this could be cited)

Reply 2: We agree with this comment. Non-stapled linear peptides composed mainly of strongly helicogenic amino acids adopt both 3₁₀- and α -helical conformations, depending on their amino acid sequences, external stimuli and aggregation. However, this is not the case for the stapled α -helical peptides described in this paper, whose side chains are intramolecularly linked with a cross-linker of the appropriate length at the appropriate positions, as already described in the "Molecular design and synthesis of stapled peptides" section in the original manuscript (page 7, line 1 from the bottom–page 8, line 1 from the top). This is due to the structural constraint imposed by the stapling. To confirm this, we have conducted additional DFT calculations for the singly stapled 3₁₀-helical **c1-Val-OH** and the doubly stapled 3₁₀-helical **dc2-Aib-OMe**. The energy-minimized structures of these 3₁₀-helical peptides and their structural parameters have been added to Supplementary Figures 4b (for **c1-Val-OH**) and 5b (for **dc2-Aib-OMe**) and the Supplementary Tables 3 and 4 (for **c1-Val-OH**) and 7 and 8 (for **dc2-Aib-OMe**) in the revised SI, respectively. According to these changes, other parts have been properly modified. As expected, the energy-minimized 3₁₀-helical structures of **c1-Val-OH** and **dc2-Aib-OMe** are 50.4 and 83.5 kJ/mol, respectively, less stable than those of the α -helical structures (Supplementary Figures 4 and 5), whereas the α -helical structure of the corresponding non-stapled analogue of **c1-Val-OH** was only 17.6 kJ/mol more stable than that of the non-stapled 3₁₀-analogue. In these stapled 3₁₀-helical structures, all piperidine rings of **c1-Val-OH** and **dc2-Aib-OMe** with a biphenyl staple are more distorted compared to those of the corresponding α -helical structures due to the structural constraint imposed by the stapling. The results have been briefly discussed in the revised manuscript (page 13, lines 6–8 and page 14, lines 2–4) and the calculation procedures have also been described in the revised SI (page S30).

The suggested reference paper has been added as reference number 42 in the revised manuscript. To incorporate this change, the following sentence "The preferential formation of the 3₁₀- or α -helix is highly dependent on the sequences of..." (page 7, lines 2–6 in the original manuscript) has been changed to "The preferential formation of the 3₁₀- or α -helix is highly dependent on the sequences of helicogenic achiral amino acid residues³⁶, such as

α -aminoisobutyric acid (Aib)³⁷, 1-aminocyclohexane-1-carboxylic acid (Ac₆C)³⁸ and 4-aminopiperidine-4-carboxylic acid (Api)³⁹, external stimuli (temperature⁴⁰ and solvent^{39,41}) and aggregation⁴², although most of...". As well, the new references (ref no. 36, 40 and 41) regarding a transition between the 3₁₀- and α -helices have been added in the revised manuscript.

Comment 3: Thermodynamically induced helicity is well established, but the resulting diastereoisomeric pairs typically interconvert rapidly; removal of the asymmetry inducing the helix leads to racemization. However, Yashima, Maeda and others have reported several examples where this is not the case: these are not peptides, but nonetheless deserve citation – especially those polymers that can be induced into a single screw sense by a chiral guest and their stabilities have a very significant dependence on solvent Eg doi/10.1002/anie.202217020

Reply 3: We agree with this comment and the suggested reference has been added as reference number 25 in the revised manuscript.

Comment 4: Helix stapling is well established to lower the rate of interconversion of helical peptides, as reported by one of the current authors in 2008. In that instance, the stapling linked residue is separated by 3 units, consistent with the formation of a 3₁₀ helix. The racemisation half life was of the order of 1 minute at room temperature.

Reply 4: We agree with this comment.

Comment 5: In the current paper, stapling is used to slow helix into conversion in similar structures, but the staple spans 7 residues, consistent with adoption of an alpha-helical structure. The resulting helices interconvert extremely slowly.

Reply 5: We agree with this comment.

Comment 6: On p9, the CD of the single stapled peptide is referred to. Alpha and 3₁₀ helices may be characterised by the relative intensity of the bands at ~208 into 210 nm. For 3₁₀ helices, the band at 220 nm can vary in sign between positive and negative (compare J. Am. Chem. Soc. 1996, 118, 11, 2744–2745 with Nature 2022, 607, 387 and discussion in E. Longo , A. Moretto , F. Formaggio and C. Toniolo , Chirality, 2011, 23 , 756 —760 and footnote 30 of Chem. Commun., 2020, 56, 12049-12052), for reasons that are still not yet clear, but nonetheless the band at 208 nm is diagnostic of screw sense. Because they have been determined in chloroform, but the CDs in this paper do not show wavelength below 220 nm, which makes the detailed discussion of conformation in solution using CD impossible (changes in screw sensible inversion of elicited is detectable, but not absolute screw sense for conformational preference). It also means that the comparison between the CDs of singly and doubly stapled peptides is not simple matter of screw sense reversal – it is possible that subtle differences between alpha and 3₁₀ preferences are at play here as well. The CD should be repeated in solvents that allow shorter wavelength determination of molar ellipticity. The band at 260 nm is presumably due to the biphenyl staple. This deserves comment.

Reply 6: Thank you very much for this important and constructive comment. Based on this comment, we have performed the CD measurements of **c1-Val-O'Bu**, **c1-Val-OH** and **dc2-Val-OH** in protic solvents with low absorption in the far-UV region (190-250 nm), such as methanol and 2,2,2-trifluoroethanol (TFE), which allowed us to obtain far-UV CD spectra of these peptides. The CD results have been added to Supplementary Figures 21a (for **c1-Val-O'Bu** and **c1-Val-OH**) and 27 (for **dc2-Val-OH**) in the revised SI. The CD signs of these peptides at 208 nm are the same as those at 224 nm, suggesting that the preferred helix sense of the stapled peptides used in this study can be assigned by the CD sign at 224 nm. Therefore, the initial assignment of their preferred helix sense using the CD sign at 224 nm is correct. Interestingly, the preferred helix sense of **c1-Val-OH** in methanol was found to be opposite to that in dichloromethane and TFE (Figure 2a(i) and Supplementary Figure 21a(iii),(iv)). This is probably due to disruption of the intramolecular hydrogen bonding of the terminal carboxy proton by methanol. In addition, the singly stapled peptides, **c1-Val-O'Bu** and **c1-Val-OH**, in both methanol and TFE displayed a CD spectral pattern similar to that of standard α -helical peptides with two maxima at around 208 and 222 nm, although the CD intensities of these stapled peptides at 222 nm were larger than those at 208 nm most likely due to the contribution of the CD signal of the biphenyl chromophore. The brief discussions of the CD spectral pattern of these peptides, the solvent-induced helicity inversion and the band at 260 nm have been added in the revised manuscript (page 9, line 5 from the bottom to page 10, line 5 and page 11, lines 3–7 and page 23, Figure caption of Fig. 4b).

Based on the newly observed solvent-induced helicity inversion, we have changed the assignment of the helical sense of **c1-Val-OH** from "*P*" and "*M*" to "*P* or *M*" in the ^1H NMR spectra measured in $(\text{CD}_3)_2\text{SO}/\text{CD}_2\text{Cl}_2$ (4/1, v/v). This change is because it is possible that dimethyl sulfoxide (DMSO), a hydrogen-bond disrupting solvent, could induce an *M*-handedness in **c1-Val-OH**, although the strong absorption of DMSO below 280 nm prevents the CD measurement to determine the preferred handedness. We had inadvertently used **c1-Val-OH** with an impurity generated by treatment of **c1-Val-O'Bu** with TFA/ CH_2Cl_2 instead of $\text{HCO}_2\text{H}/\text{CH}_2\text{Cl}_2$ for 1D and 2D ^1H NMR measurements in $(\text{CD}_3)_2\text{SO}/\text{CD}_2\text{Cl}_2$ (4/1, v/v) (Supplementary Figures 6–10 in the original SI). Therefore, these measurements have been repeated with the pure **c1-Val-OH** and the old spectra have been replaced with the new ones (Supplementary Figures 8–12 in the revised SI), which did not change the conclusion except for the *P/M* assignment.

Comment 7: Barrier determinations also appear to have been done in CH_2Cl_2 , which is not at all biomimetic, although it can be used as a model for phospholipid membranes. Similar determination should be carried out for protic solvents, especially given the suggestion from the authors that these molecules have the potential to be used in the context of medicinal chemistry.

Reply 7: As described in the reply to this reviewer's comment 6, we found that the preferred helix sense of **c1-Val-OH** in methanol is opposite to that in CH_2Cl_2 and TFE. Hence, a solution of **c1-Val-OH** in TFE was 50 times diluted with methanol at 45, 35, 25 and 15 °C and the time-dependent CD intensity changes were monitored immediately after the dilution. The time-dependent CD intensity changes and the kinetic and thermodynamic activation parameters have been added to Supplementary Figure 21b-d and Supplementary

Table 12, respectively, in the revised SI. The activation energy (E_a) value of **c1-Val-OH** in methanol/TFE (49/1, v/v) was estimated to be 89.7 kJ/mol, which is approximately 15 kJ/mol lower than that in CH₂Cl₂. A similar tendency was also observed for **dc2-Val-OH** as indicated by the kinetic and thermodynamic data obtained from the time-dependent CD intensity changes of the as-obtained **dc2-Val-OH** in *n*-butanol instead of the low boiling-point methanol at 110, 100, 90 and 80 °C (Supplementary Figure 28 and Supplementary Table 13 in the revised SI). A similar protic solvent-induced acceleration of the *P/M* interconversion is also known (Kubasik, M. & Blom, A. *ChemBioChem* **6**, 1187-1190, (2005), which has been added as reference number 53 in the revised manuscript). The solvent effect on the *P/M* interconversion rate has been briefly discussed in the revised manuscript (page 18, lines 3–9 for **c1-Val-OH** and page 22, lines 10–13 for **dc2-Val-OH**). The molecules in this study are insoluble in water, which prevents their study in this solvent.

Comment 8: Likewise, the NMR experiments in indicate alpha-helical structures, but given the solvent dependence of 310 and alpha-helical preferences it should be explored whether this is consistent across other more protic solvents.

Reply 8: According to this comment, we have tried to perform ¹H NMR measurements of **c1-Val-OH** in a CD₃OH/CD₂Cl₂ mixture (5/2, v/v) instead of pure CD₃OH, because of its low solubility in pure CD₃OH. As a result, no significant difference was observed among the ¹H NMR spectra in a CD₃OH/CD₂Cl₂ mixture (5/2, v/v), CD₂Cl₂ and a (CD₃)₂SO/CD₂Cl₂ mixture (4/1, v/v) (Review-Only Supplementary Figure 1). This is consistent with the small CD spectral changes of the as-obtained **dc2-Val-OH** around 222 nm upon changing solvents (Figure 4b and Supplementary Figure 27 in the revised SI), in which the *P/M* molar ratios in each solvent were assumed to be identical. This suggests that the peptides did not undergo a solvent-induced 3₁₀/α-helix transition, which induces a significant change in the CD intensity around 222 nm (Refs. 40–42 and 44 in the revised manuscript). As already described in "Reply 2" to this reviewer, the DFT calculations revealed that the α-helical structures of **c1-Val-OH** and **dc2-Aib-OMe** are significantly more stable than the corresponding 3₁₀-helical structures. The biphenyl stapling of the 3₁₀-helices caused a significant structural strain at the A_{pi} side chains, thus exclusively inducing the α-helical structure in these stapled peptides.

Comment 9: Overall these are interesting results, and the degree to which stapling slows the rate of helical inversion is significant, though perhaps unsurprising given the degree to which the peptide is rigidified by two staples. However there are many questions over the assignment of conformation that need clarifying before the work becomes publishable.

Reply 9: We again thank the reviewer for his/her careful evaluation of our work. We have revised our manuscript and Supplementary Information according to the valuable comments and we now believe that the revised manuscript has been improved and is ready for publication in *Nature Communications* as an Article.

Response to Reviewer #2:

Thank you very much for your kind review and valuable comments for our manuscript. According to the suggestions, we have revised our manuscript as follows. The changes are also noted in the revised manuscript and Supplementary Information (SI) using red text.

Comment 1: This is a very interesting contribution in the field of biotic foldamers done mainly by Naoki Ousaka who has a large experience in the field. Although the title suggest that the manuscript will be focused on achiral stapled foldamers, there is also a large part of the article, at least 70%, dealing with conformational communication along the foldamer chain that is derivatized at the C terminus with a chiral val residue. The authors clearly show how the pH affects to the conformational composition at the C-terminus chiral val residue, and how this changes affect to the screw sense excess of the foldamer. From these studies, they clearly show that while the single stapled foldamer is still dynamic, the double stapled one is quasi-static. There is no info about this part of the work neither in the abstract or title and can be missed during literature search.

Reply 1: Thank you very much for this constructive comment. According to the comment, we have properly modified the title and abstract to cover important points of our manuscript. The title has been changed to "Stapling strategy for slowing helicity interconversion of α -helical peptides and isolating chiral auxiliary-free one-handed forms".

Comment 2: The last part of the work deals with the isolation of both P and M helical structures from a completely achiral foldamer due to the double stapling strategy, which practically freeze the two P and M macromolecular helical conformers, making possible to isolate them and keep it stable in solution for a while.

Reply 2: We agree with this comment.

Comment 3: I think that the research is very interesting for the scientific community working in biological chemistry, materials science, organic and inorganic chemistry (foldamers, metallofoldamers, peptides, DNA, helical covalent and supramolecular polymers). The work has been done carefully, all the data is clearly shown, and discussion is easy to follow.

Reply 3: We appreciate this positive and encouraging comment.

Response to Reviewer #3:

Thank you very much for your review and valuable comments for our manuscript. According to the suggestions, we have revised our manuscript as follows. The changes are also noted in the revised manuscript and Supplementary Information (SI) with red color.

Comment 1: The manuscript by N. Ousaka and coworkers deals with the solution synthesis of six peptides, of which two made only of achiral residues. Most of the text

refers to the Val-containing compounds termed c1-Val and dc2Val, however, therefore the title, abstract and conclusions do not reflect the results.

Reply 1: Thank you very much for this constructive comment. According to the comment, we have changed the title, abstract and conclusions to better emphasize the important points of our manuscript.

Comment 2: The description of a single-handed alpha-helix made of only achiral residues not obtained by any kind of chiral interactions is novel and deserves publication, but unfortunately this is not the case. There are several flaws in the present version of the manuscript:

Reply 2: According to this comment, we have revised our manuscript as follows.

Comment 2-1-1: 1) Data on the peptides c1-Val-OH/OtBu and dc2-Val-OH/OtBu are not pertinent to the present article and should be removed totally. There are several studies in the literature on peptides containing just one chiral residue that adopt a single-handed helix.

Reply 2-1-1: We decided to include the data on the peptides **c1-Val-OH/O'Bu** and **dc2-Val-OH**, because these data are important pieces of this study. In this study, we focused on development of well-defined stapled α -helical peptides with slow and/or almost no *P/M* interconversion. The introduction of the C-terminal unprotected Val-OH residue at the C-terminal end of the stapled achiral peptides was necessary to facilitate the monitoring of the kinetics for *P/M* interconversion by CD measurements, especially for the singly stapled **c1-Val-OH**, of which *P/M* interconversion took place on a time scale of minutes. This time scale is much shorter than the chromatographic time scale, making kinetic studies of *P/M* interconversion difficult or inaccurate. Therefore, the acid/base- or solvent-induced helicity inversion system (newly added in the revised manuscript) used in this study is useful for unambiguous determination of kinetic parameters for the *P/M* interconversion.

Comment 2-1-2: By the way, there are no clear data in support of those Val-containing peptides adopting a alpha-helical conformation, since CD has been performed in dichloromethane and the two maxima at 208 and 195 nm are not detectable. Both their position and relative intensity (especially $[\theta_{222}]/[\theta_{208}]$ ratio) could have given indications on the helix being alpha or 3-10.

Reply 2-1-2: Thank you very much for this important comment. A similar comment was also raised by Reviewer #1 (Comment 6). Please also see our "Reply 6" to the Reviewer #1. According to this comment, we have performed the CD measurements of **c1-Val-O'Bu**, **c1-Val-OH** and **dc2-Val-OH** in protic solvents, such as methanol and 2,2,2-trifluoroethanol (TFE). These peptides in both methanol and TFE displayed a CD spectral pattern similar to that of standard α -helical peptides with two maxima at around 208 and 222 nm, although the CD intensities of these stapled peptides at 222 nm were larger than those at 208 nm most likely due to the contribution of the CD signal of the biphenyl chromophore. The brief discussion of the CD spectral pattern of these peptides has been added in the revised manuscript (page 9, line 5 from the bottom to page 10, line 5 and page 11, lines 3–7 and

page 23, Figure caption of Fig. 4b). Additional explanations described in our "Reply 2" and "Reply 8" to the Reviewer #1 also support the α -helix formation of these stapled peptides.

Comment 2-1-3: The authors claim (p.12, line 8) that the presence of all detectable sequential NH-NH correlations in the NOESY spectrum is diagnostic of a α -helix, but this observation is compatible with both kind of helices (see Wüthrich's book).

Reply 2-1-3: We did not specify that the presence of all detectable sequential NH-NH correlations in the NOESY spectrum is diagnostic of an α -helix, but mentioned that this observation is consistent with the energy-minimized α -helical **c1-Val-OH** obtained by the DFT calculation (Fig. 3a).

Comment 2-1-4: By the way, apparently the authors did not produce the build-up curve to determine the mixing time for their NOESY, therefore there might be spin-diffusion, especially since the peptide concentration is quite high.

Reply 2-1-4: According to this comment, we have repeated the ^1H - ^1H NOESY measurement of **c1-Val-OH** in $(\text{CD}_3)_2\text{SO}/\text{CD}_2\text{Cl}_2$ (4/1, v/v) at various mixing times. However, no significant difference was observed when the NOESY measurements were conducted with mixing times around 500 msec.

Comment 3: 2) All studies on the Val-containing peptides presented here should have been performed on the Aib-containing ones instead. All paragraphs: "Conformational analysis of singly and doubly stapled peptides" (p.9) - M/P ratio determination by ^1H -NMR included; "Acid/base-triggered reversible helicity inversion" (p.15) and related kinetics and thermodynamic parameters should be rewritten with data collected on the achiral peptide(s) dc2-Aib-OH/OMe.

Reply 3: As already mentioned in "Reply 2-1-1" to this reviewer, we would not like to perform these changes raised by this reviewer. In principle, it is impossible to determine *M/P* molar ratios of singly and doubly stapled achiral peptides having no chiral residue such as a Val residue, because the magnetic environments of such enantiomeric helices are identical to each other. In addition, stapled achiral peptides having the C-terminal unprotected Aib-OH residue do not undergo an acid/base-triggered change in their *M/P* molar ratio upon addition of the organic bases used in this study, because these bases are achiral.

Comment 4-1: There are also other, important issues to be tackled:

i) Figure 2, p.11 and par. "Acid/base-triggered reversible helicity inversion", p.15. There must be indications on the apparent pH of the solutions used. The differences between bulky, CHIRAL (except for A3), cyclic amines and linear, achiral ones seem to imply that it is not really the basic pH that matters, but rather the association with the counterion, which acts as an 'auxiliary', with chirality on the nitrogen. "helicity inversion triggered by association with a counterion" seems a better choice for this phenomenon.

Reply 4-1: Thank you for this valuable comment. However, we are concerned that this reviewer may have misunderstood that we used aqueous solutions and chiral bases for spectroscopic measurements. We used non-aqueous solutions and achiral bases, such as TBD, DBU, an amidine (**A3**) and aliphatic amines (ⁿBuNH₂, NHEt₂ and NEt₃) as listed in Fig. 2c. The determination of the apparent pH of the organic solvents used in this study is not possible, instead, the pK_a values of TBD, DBU and NEt₃ (in acetonitrile) are reported to be 26.03, 24.34 and 18.82, respectively (Leito, I. *et al*, *J. Org. Chem.* **70**, 1019-1028 (2005)).

Comment 4-2: ii) Figure 2a: there's a band centered at about 270 nm, which is used afterwards to make several assumptions. Such a dichroic signal for helical peptides is usually due to the presence of aromatic groups experiencing environmental chirality (from the helix, other chiral centers nearby, etc). This means: 1. also the band at about 224nm can have a contribution by aromatic groups, therefore its intensity cannot be safely attributed to the conformation alone. 2. Since Val-OtBu does not show the same band, but the peptide backbone should be the same (Fig. 1c) then something is missing: either the presence of the additional H-bond c1-Val-OH strengthen the helix to the point that the aromatic groups becomes optically active and this does not happen for c1-Val-OtBu (although that 'aromatic' band is present also for dc2-Aib-OMe!), or all c1-Val-OH spectra (and dc2-Aib-OMe) were acquired in the presence of an aromatic contaminant able to closely interact with the peptide itself and not present in the solutions of c1-Val-OtBu. An explanation on this point is crucial, since - as already pointed out - the same band is present also for dc2-Aib-OMe. Moreover, the reasons why such a band has been chosen for the following studies are not clear and should be explained.

Reply 4-2: Thank you very much for this valuable comment. The normalized CD spectra of **c1-Val-OH** and **c1-Val-O^tBu** and the as-obtained and annealed **dc2-Val-OH** are pseudo-mirror images of each other (Review-Only Supplementary Fig. 2), indicating that the CD band at around 260 nm and around 224 nm are completely linked to each other. The CD intensity changes of **dc2-Val-OH** and **dc2-Aib-OMe** were monitored at 272 nm instead of 224 nm, because the use of 1,1,2,2-tetrachloroethane as a high-boiling solvent prevents the CD measurement below 250 nm. This explanation has been added in the revised SI (page S63, lines 7–9).

Comment 4-3: iii) Paragraph "Optical resolution of rac-dc2-Aib-OMe and its racemization kinetic", p.24-25. This is the core of the article, but the kinetics and thermodynamic parameters are not reported in the main article, and how they had been obtained not clearly explained. Once all the Val-peptides are removed from the article, the corresponding data for Aib-peptides should be added. The NMR (?) studies from which the M/P ratio and interconversion rate were extrapolated are not explained nor reported, but should be carried out if not yet done. Fig 24, p. S61 and Fig. 25, p. S62, refer to CD studies on the aromatic band at about 270nm and should be supported with the corresponding NMR study. Fig 24 at p. S61 is not sound, since the timeframe of the time-dependence CD is different among the temperatures. In particular, 10 minutes at 100°C is too short.

Reply 4-3: The kinetics and thermodynamic parameters for **dc2-Aib-OMe** were already summarized in Tables 4 and 2, respectively, in the original main article. In addition, the detailed procedure for determining these parameters was already in the original SI (section 13, page S69).

As already described in our "Reply 3" to this reviewer, it is impossible to determine *M/P* molar ratios of **dc2-Aib-OMe** having no chiral residue such as a Val residue by NMR measurements, because the magnetic environments of the enantiomeric helices are identical to each other. Thus, we determined the *M/P* molar ratio changes and the *P/M* interconversion rates by monitoring the CD intensity changes at 272 nm.

The time-dependent CD intensity changes of **dc2-Aib-OMe** at 100 °C for 13.6 min provided the rate constant (k_{rac}) with a rather small deviation ($\pm 2\%$). Thus, the monitoring time of approximately 10 minutes is sufficient to estimate the accurate rate constant under this condition. Nevertheless, the time frame of the time-dependent CD changes at each temperature in Supplementary Fig. 24a in the original SI has been changed to be the same as the others (Supplementary Fig. 29a in the revised SI).

Comment 4-4: iv) the reasons why dichloromethane is used for the CD study must be included. Why not methanol or other solvents which are more transparent and show all the CD bands?

Reply 4-4: The use of CH₂Cl₂ for the CD study is mainly due to two reasons: 1) The stapled peptides used in this study are much more soluble in CH₂Cl₂ than in methanol. 2) The association constant of **c1-Val-OH** with TBD in CH₂Cl₂ containing only 5% vol. methanol was estimated to be $(1.57 \pm 0.46) \times 10^5 \text{ M}^{-1}$ (Review-Only Supplementary Fig. 3), which is approximately an order of magnitude lower than that in CH₂Cl₂.

Comment 5: Some minor points: -p.6, Figure 1c,d: R, R' at the C-terminus would avoid confusion. A table with the sequences would be helpful.

Reply 5: Thank you for this comment; we have modified Figure 1 accordingly.

Comment 6: -final purity of the peptides should be provided somewhere.

Reply 6: We have performed HPLC analyses of **c1-Val-O'Bu**, **c1-Val-OH**, **dc2-Aib-OMe** and **dc2-Val-OH**. The HPLC chromatograms of these peptides have been added to Supplementary Fig. 3 in the revised SI.

Comment 7: - p. S14, lines 6-8. Washings with a CHCl₃/MeOH/EtOAc mixture is a non-standard procedure in solution-phase peptide synthesis, perhaps it should be further explained to allow for reproducibility.

Reply 7: The volume ratio of a CHCl₃/MeOH/EtOAc mixture (ca. 9/1/50, v/v/v) was already provided in the original SI.

Comment 8: - p. S28, supplementary figure 2. The complete (50-2500Da) mass spectra should be provided.

Reply 8: The full region ESI-TOF mass spectra have now been added to Supplementary Fig. 2 in the revised SI.

Comment 9: - p. S53, suppl. figure 18. The authors should state how they performed the M/P assignments. The 2D NMR spectrum of the same peptide c1-Val-OH, reported at p. S41, Figure 8 was acquired in another solvent and could not have been of help.

Reply 9: The *M/P* assignments in the ^1H NMR spectra of **c1-Val-OH** in the absence and presence of 1.2 equiv of TBD shown in Supplementary Fig. 18 (in the original SI) have been done on the basis of the CD measurement results shown in Fig. 2a(i),(ii).

We are grateful for the reviewer's detailed suggestions and we now believe that the revised manuscript has been improved and is ready for publication in *Nature Communications*.

REVIEWER COMMENTS

Reviewer #1 (Remarks to the Author):

The authors have made a very careful and thoughtful revision of the paper taking into account all of the concerns of the referees, and I feel that it has been significantly improved by these changes.

I would therefore be happy to recommend acceptance in its current form.

Reviewer #3 (Remarks to the Author):

The authors addressed the reviewers' concern. The title, abstract and conclusions were amended and now reflect the article content. The HPLC profiles of the peptides (Supplementary figure 3) were added and they seem to reflect a rather poor purity for some peptides. The high resolution ESI-MS spectra are not reported. HRMS (containing 4 digits) is required to unambiguously assign multiple charges, especially when multiple cations are claimed (e.g., $[M+3Na]^{3+}$). VT NMR with 2 NHs moving means 3-10 helix. α -helices display 3 NHs sensitive to the environment. A clear experimental indication of the presence of a α -helix seems missing. Comment 9 on the old figure S18 (now S20) refers to proton assignments. It would be helpful to state on the caption to figure S20 how the protons of the two helical screwsenses were assigned. Perhaps 2D NMR under the same experimental conditions were acquired, to decide that the red protons are all from one helix and the blue ones are from the other?

Response to Reviewer #3:

Thank you very much for your review and valuable comments for our revised manuscript. According to the suggestions, we have thoroughly revised our manuscript as follows. The changes are also indicated in the revised Supplementary Information (SI) with a red color text.

Comment 1: The authors addressed the reviewers' concern. The title, abstract and conclusions were amended and now reflect the article content.

Reply 1: We agree with this comment.

Comment 2: The HPLC profiles of the peptides (Supplementary figure 3) were added and they seem to reflect a rather poor purity for some peptides.

Reply 2: We do not agree with this comment. The reverse-phase HPLC chromatograms of the peptides showed no peaks corresponding to impurities, except for the solvent peak (CH₂Cl₂ used as the injection solvent) with a retention time of about 1.5 min (Supplementary Fig. 3 in the previously revised SI). In these chromatograms, the peptides having the L-Val residue at the C-terminus (**c1-Val-O'Bu**, **c1-Val-OH** and **dc2-Val-OH**) displayed two peaks due to their diastereomeric (*P*)- and (*M*)-helices, whereas **dc2-Aib-OMe** without any chiral auxiliaries showed a single peak (retention times of the enantiomeric (*P*)- and (*M*)-helices are identical.). These observations are consistent with their ¹H NMR spectra, in which **c1-Val-O'Bu**, **c1-Val-OH** and **dc2-Val-OH** showed two sets of signals and **dc2-Aib-OMe** exhibited a single set of signals (Fig. 4a and Supplementary Figs. 16 and 17b,c).

Comment 3: The high resolution ESI-MS spectra are not reported. HRMS (containing 4 digits) is required to unambiguously assign multiple charges, especially when multiple cations are claimed (e.g., [M+3Na]³⁺).

Reply 3: According to this comment, we have added HRMS values with 4 decimal digits for monovalent peaks in Supplementary Fig. 2. The multiple charge peaks (e.g., [2**dc2-Val-OH**+3Na]³⁺) shown in the ESI-TOF mass spectra of **c1-Val-O'Bu**, **c1-Val-OH**, **dc2-Aib-OMe** and **dc2-Val-OH** were already unambiguously assigned (Review-Only Supplementary Fig. 1).

Comment 4: VT NMR with 2 NHs moving means 3-10 helix. α -helices display 3 NHs sensitive to the environment. A clear experimental indication of the presence of a α -helix seems missing.

Reply 4: As already described in the previously revised manuscript (page 13, lines 1–6), the DFT study of the singly stapled α -helical **c1-Val-OH** revealed that only the amide N(1)H and N(2)H protons are exposed to the solvent (the number in parentheses represents the residue number from the N-terminus). This is because the N-terminal Api(1) residue seemed to adopt a ₃₁₀-helix-like conformation to maximize the number of the

intramolecular hydrogen bonds at the N-terminal region (Supplementary Table 2). A similar hydrogen bonding pattern was observed for the doubly stapled **dc2-Aib-OMe**, as supported by the DFT calculation and single crystal X-ray structure of its analog **dc2-Aib-OH** (Supplementary Tables 6 and 11).

Comment 5-1: Comment 9 on the old figure S18 (now S20) refers to proton assignments. It would be helpful to state on the caption to figure S20 how the protons of the two helical screw senses were assigned.

Reply 5-1: Thank you for this constructive comment. According to the comment, the following sentence has been added to the caption of Supplementary Fig. 20: "The *P/M* assignments were made based on the CD measurement results shown in Fig. 2a(i),(ii)."

Comment 5-2: Perhaps 2D NMR under the same experimental conditions were acquired, to decide that the red protons are all from one helix and the blue ones are from the other?

Reply 5-2: We did not perform 2D NMR measurements of **c1-Val-OH** in the absence and presence of TBD in CD₂Cl₂, because the *P/M* assignments were made based on the CD measurement results shown in Fig. 2a(i),(ii) and the protons of the (*P*)- and (*M*)-helices could be easily assigned based on their integral ratios. It should be noted that the *P/M* integral ratios of **c1-Val-OH** in the absence and presence of TBD were in excellent agreement with the CD measurement results.

REVIEWERS' COMMENTS

Reviewer #3 (Remarks to the Author):

The authors answered the comments.